# Molecular mechanism of β-arrestin-2 pre-activation by phosphatidylinositol 4,5-bisphosphate

Kiae Kim 🆔 & Ka Young Chung 🆔 ✉

## Abstract

Phosphorylated residues of G protein-coupled receptors bind to the N-domain of arrestin, resulting in the release of its C-terminus. This induces further allosteric conformational changes, such as polar core disruption, alteration of interdomain loops, and domain rotation, which transform arrestins into the receptor-activated state. It is widely accepted that arrestin activation occurs by conformational changes propagated from the N- to the C-domain. However, recent studies have revealed that binding of phosphatidylinositol 4,5-bisphosphate (PIP$_2$) to the C-domain transforms arrestins into a pre-active state. Here, we aimed to elucidate the mechanisms underlying PIP$_2$-induced arrestin pre-activation. We compare the conformational changes of β-arrestin-2 upon binding of PIP$_2$ or phosphorylated C-tail peptide of vasopressin receptor type 2 using hydrogen/deuterium exchange mass spectrometry (HDX-MS). Introducing point mutations on the potential routes of the allosteric conformational changes and analyzing these mutant constructs with HDX-MS reveals that PIP$_2$-binding at the C-domain affects the back loop, which destabilizes the gate loop and βXX to transform β-arrestin-2 into the pre-active state.

Keywords Arrestin; Phosphatidylinositol 4,5-bisphosphate; Structure; HDX-MS
Subject Categories Membranes & Trafficking; Signal Transduction; Structural Biology

## Introduction

Arrestins, a protein family regulating G protein-coupled receptor (GPCR) signaling, have four distinct members in mammals (arrestin-1–4) (Benovic et al, 1987; Lohse et al, 1990). Arrestin-1 and -4 are visual system-specific, while arrestin-2 and -3 (β-arrestin-1 [βarr1] and 2 [βarr2]) are widely expressed (Lohse and Hoffmann, 2014). They desensitize and internalize agonist-activated phosphorylated GPCRs (Benovic et al, 1987) and regulate other signaling pathways (Coffa et al, 2011; Park et al, 2019; Perry-Hauser et al, 2022; Perry et al, 2019; Qu et al, 2021a; Smith and Rajagopal, 2016; Srivastava et al, 2015). Understanding how arrestins are activated at the structural and molecular level is crucial for the development of drugs targeting GPCRs or related pathways.

Previous studies revealed arrestin structures in basal and receptor-bound active states (Chen et al, 2023b; Hirsch et al, 1999; Huang et al, 2020b; Lee et al, 2020; Mayer et al, 2019; Park et al, 2019; Shukla et al, 2014; Staus et al, 2020; Yang et al, 2015; Yun et al, 2015; Zhou et al, 2017). Arrestins consist of N- and C-domains with a seven-stranded β sandwich in each domain (Fig. 1A). The basal state is stabilized by the interaction between the C-tail, more precisely βXX, and N-domain (Fig. 1A, purple circle) and the polar core formed by ionic interactions between residues within the gate loop, βIII, βX, and C-tail (Fig. 1A, orange circle).

Binding of the phosphorylated GPCR (Fig. 1B, green) at the N-domain transforms arrestins into the active state by releasing βXX, disrupting the polar core, and affecting the conformation of the loops between the N- and C-domains, and altering the relative interdomain orientation (Fig. 1B). Although these conformational changes are the "canonical" changes of the receptor-activated arrestins, the degree of these changes can vary depending on the receptor types and phosphorylation patterns, resulting in different arrestin active states and functional outcomes (Kaya et al, 2020; Latorraca et al, 2020; Maharana et al, 2023b; Mayer et al, 2019; Yang et al, 2015; Zhou et al, 2017).

In recent years, plasma membrane components including phosphatidylinositol 4,5-bisphosphate (PIP$_2$) have been implicated in βarr activation (Eichel et al, 2018; Grimes et al, 2023; Huang et al, 2020b; Janetzko et al, 2022; Kang et al, 2015; Zhai et al, 2023). High-resolution structures of GPCR-βarr complexes show βarr's C-domain contacting lipids or detergents (Chen et al, 2023b; Staus et al, 2020), which facilitates GPCR-βarr complex formation (Lally et al, 2017; Zhou et al, 2017). The involvement of PIP$_2$ in βarr activation has been extensively suggested. With the assistance of PIP$_2$, βarr can become "catalytically activated" (i.e., active without receptor binding) (Eichel et al, 2018). A subsequent study proposed that PIP$_2$-binding is necessary for certain GPCR-βarr interactions and that PIP$_2$ promotes βarr activation (Janetzko et al, 2022). Notably, the cryo-electron microscopy structures showed PIP$_2$-binding at the C-domain of βarr1 in the neurotensin receptor 1 (NTSR1)-βarr1 and glucagon receptor (GCGR)-βarr1 complexes (Fig. 1C) (Chen et al, 2023b; Huang et al, 2020b). However, the precise structural mechanism by which PIP$_2$ promotes arrestin activation remains unclear.

Nevertheless, only few studies examined PIP$_2$-induced βarr conformational changes by labeling specific residues with a fluorophore or $^{19}$F (Janetzko et al, 2022; Zhai et al, 2023). Here, we investigated PIP$_2$-induced arrestin activation mechanism using hydrogen-deuterium exchange mass spectrometry (HDX-MS). HDX-MS monitors the exchange between the amide hydrogen in the protein and deuterium in the solvent, providing information

School of Pharmacy, Sungkyunkwan University, 2066 Seobu-ro, Jangan-gu, Suwon 16419, Republic of Korea. ✉E-mail: kychung2@skku.edu

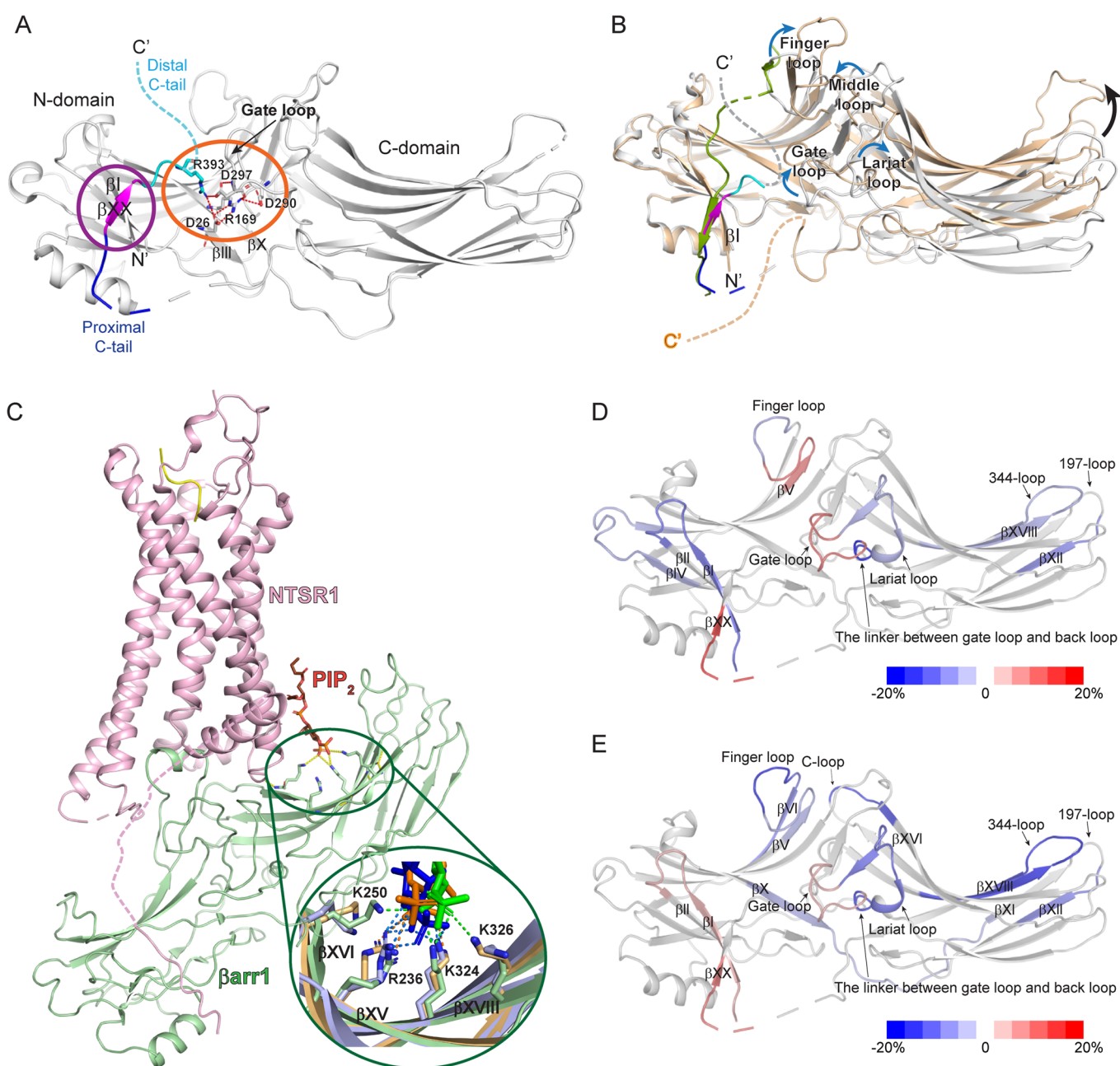

about the protein conformational dynamics (Bai et al, 1993; Mayne, 2016). We compared the conformational dynamics of the PIP$_2$-induced and the phosphorylated C-tail peptide of the vasopressin receptor type 2 (V2Rpp)-induced active states of βarr2 and explored activation mechanisms through mutational studies.

## Results and discussion

### Conformational changes of βarr2 upon PIP$_2$-binding

To investigate PIP$_2$-induced conformational changes of βarr2, purified βarr2 was incubated with water-soluble PIP$_2$ (150 μM) as

described in the Methods. Subsequently, deuterium exchange was initiated on ice for various durations (10, 100, 1000, and 10,000 s). The peptic peptides used for the HDX-MS analyses are shown in Fig. EV1, and the HDX-MS data analyzed in the present study are summarized in Dataset EV1. To compare the PIP$_2$-induced structural changes with phosphorylated GPCR-induced changes, we also examined the effects of V2Rpp (500 μM), a well-established model system for understanding βarr interactions with phosphorylated receptor C-tails (Fig. 1B) (Latorraca et al, 2020; Mayer et al, 2019; Shukla et al, 2013b; Yang et al, 2015).

V2Rpp induced higher HDX in the N-terminal half of the finger loop, gate loop, and proximal C-tail through βXX (Figs. 1D and EV2, peptides 62–69, 292–302, and 382–389), and lower HDX in a few

◀ **Figure 1. Structures of βarr in various states and HDX-MS profile changes upon the binding of V2Rpp or PIP₂ to βarr2.**

(A) Structure of βarr1 in the basal state (PDB: 1G4R) (Data ref: Han et al, 2001a; Han et al, 2001b). The basal state βarr1 is colored gray with the C-terminus colored blue (proximal C-tail), magenta (βXX), and cyan (distal C-tail). Unresolved regions are indicated by dotted lines. The interaction between βXX and the residues in the N-domain is indicated in the purple circle, and the polar core is denoted in the orange circle. Residues that are involved in the polar core formation are shown as sticks. (B) Comparison of the structure of βarr1 in basal (PDB: 1G4R) (Data ref: Han et al, 2001a; Han et al, 2001b) and V2Rpp-bound (PDB: 4JQI) (Data ref: Shukla et al, 2013a; Shukla et al, 2013b) states. V2Rpp-bound βarr1 is colored light orange and V2Rpp is colored green. The color codes for the basal state of βarr1 are same as those of (A). The conformational changes of the loop regions are shown with blue arrows, and the domain rotation is indicated with a black arrow. (C) Structure of the NTSR1-βarr1 complex (PDB: 6UP7) (Data ref: Huang et al, 2020a; Huang et al, 2020b). NTSR1 is colored light pink, and βarr1 is colored light green. PIP₂ is indicated with orange sticks. The residues that interact with PIP₂ are shown as sticks. In the enlarged green circle, various modes of interaction between βarr1 and PIP₂ are shown; PIP₂ in the NTSR1-βarr1 complex (PDB: 6UP7) (Data ref: Huang et al, 2020a; Huang et al, 2020b) is colored orange, the interacting residues in βarr1 are colored light orange, and the ionic interactions between PIP₂ and βarr1 are shown as green dotted lines; PIP₂ in the GCGR1-βarr1 complexes (PDB: 8JRU and 8JRV) (Data ref: Chen et al, 2023a; Chen et al, 2023b) is colored blue or green, the interacting residues in βarr1 are colored light blue or light green, and the ionic interactions between PIP₂ and βarr1 are shown as blue or orange dotted lines respectively. (D) HDX-MS profile comparison between the apo and V2Rpp-bound βarr2. The HDX-level differences (i.e., HDX levels of apo βarr2–HDX levels of V2Rpp-bound βarr2) are color-coded on the basal state structure of βarr2 (PDB: 3P2D) (Data ref: Zhan et al, 2011a; Zhan et al, 2011b). Results were derived from three independent experiments. (E) HDX-MS profile comparison between the basal and PIP₂-bound βarr2. The HDX-level differences (i.e., HDX levels of apo βarr2–HDX levels of PIP₂-bound βarr2) are color-coded on the basal state structure of βarr2 (PDB: 3P2D) (Data ref: Zhan et al, 2011a; Zhan et al, 2011b). Results were derived from three independent experiments (biological). The color-coded HDX-level differences are based on the maximum differences at any D₂O incubation time point. The detailed HDX-MS data are summarized in Dataset EV1 and Fig. EV2.

regions within the N-domain (βI, βIV through βV, and C-terminal half of the finger loop; peptides 1–19, 41–55, and 70–76), domain interfaces (the lariat loop and the linker between the gate loop and back loop; peptides 281–291 and 303–306), and a few regions within the C-domain (197-loop and βXVIII; peptides 195–201 and 324–338) (Figs. 1D and EV2).

These HDX-MS data well-reflected the known V2Rpp-induced conformational changes of βarr2. V2Rpp (Fig. 1B, green) interacts at the N-domain groove and near βI. Thus, lower HDX levels of the V2Rpp-bound βarr2 at the N-domain (specifically, βI and C-terminal half of the finger loop) probe the V2Rpp-binding in these regions. In addition, V2Rpp-induced higher HDX levels in the gate loop and proximal C-tail through βXX indicate conformational changes resulting from βXX release and polar core disruption. The HDX-level changes at the domain interfaces suggest conformational changes in the loop regions at the domain interfaces and/or domain rotation upon V2Rpp-binding, and the changes at the C-domain may reflect long-range allosteric conformational changes transmitted from the N-domain V2Rpp-binding site.

As HDX-MS analysis effectively probed the V2Rpp-induced activation of βarr2, we sought to analyze the PIP₂-induced conformational changes of βarr2. Based on the NTSR1-βarr1 and GCGR-βarr1 complex structures, PIP₂ can interact with positively charged residues at βXV (R236 in the βarr1 sequence), βXVI (K250 in the βarr1 sequence), and βXVIII (K324 and K326 in the βarr1 sequence) (Fig. 1C, inlet). The HDX-MS analysis revealed that the HDX levels at βXVIII become lower upon co-incubation with PIP₂ (Figs. 1E and EV2, peptide 324–338), implying the binding of PIP₂ to βarr2. However, the HDX levels of the peptides covering βXV (Figs. 1E and EV2, peptide 219–239) and βXVI (Figs. 1E and EV2, peptide 251–258) were not affected. This may be due to three reasons. First, as HDX monitors the buffer exposure of the amide hydrogens at the peptide backbone, HDX levels could not be affected if the binding occurs through the charge–charge interaction mediated by the amino acid side chains without altering the peptide backbone conformation. Second, the PIP₂-interacting residues may differ slightly between the receptor-bound (i.e., the NTSR1-βarr1 and GCGR-βarr1 complexes) and unbound states (i.e., current study). Even in the receptor-bound states, PIP₂ interacted differently between the NTSR1-bound and GCGR-bound states

(Fig. 1C, inlet). Third, the reported βarr structures with PIP₂ are βarr1 structures (Fig. 1C), but in this study, we analyzed the conformation of βarr2. Therefore, the differences may stem from variations between these subtypes. Nevertheless, the HDX-MS data indicate that PIP₂ interacts at the positively charged region within the C-domain of βarr2.

Interestingly, PIP₂ induced higher HDX levels at βI, gate loop, and proximal C-tail through βXX (Figs. 1E and EV2, peptides 1–19, 292–302, and 382–389), which is the canonical feature of the βarr activation (i.e., βXX release and polar core disruption) (Fig. 1B) (Kim et al, 2015; Shukla et al, 2014; Yun et al, 2015). Of note, the HDX levels of the PIP₂-bound state at the gate loop and proximal C-tail through βXX were still lower than the V2Rpp-bound state (Fig. EV2, peptides 292–302 and 382–389), which suggests that the PIP₂-bound state is not as fully active as the V2Rpp-bound state. Thus, these results suggest that the binding of PIP₂ destabilizes the gate loop and the interaction of βXX at the N-domain, which may transform βarr2 more amenable to be activated (i.e., pre-active state).

PIP₂-induced HDX-level changes were also evident at the finger loop, βVI, βX through βXI, 197-loop, C-loop, lariat loop, and the linker between the gate loop and back loop (Figs. 1E and EV2, peptides 62–69, 70–76, 75–81, 168–186, 195–201, 246–250, 281–291, and 303–306). Although most of these regions were also affected by V2Rpp-binding, the HDX-MS profiles at the finger loop and its extension (i.e., βVI) (Figs. 1E and EV2, peptides 62–69, 70–76, and 75–81) and the lariat and gate loops (Figs. 1E and EV2, peptides 281–291 and 292–302) differed between the V2Rpp- and PIP₂-bound states, suggesting that these regions adopt different conformations between V2Rpp- and PIP₂-bound states. Furthermore, βX through βXI and the C-loop were affected by PIP₂, but not by V2Rpp (Figs. 1E and EV2, peptides 168–186 and 246–250).

## Distal C-tail of βarr2 is not involved in PIP₂-induced pre-activation

The HDX-MS data suggest that the interaction of PIP₂ at the C-domain affects the conformational dynamics of βI, gate loop, and βXX (Fig. 1E) potentially through the allosteric transmission of the conformational changes from the C-domain to the gate loop and βXX. Thus, we sought to understand the routes for the allosteric

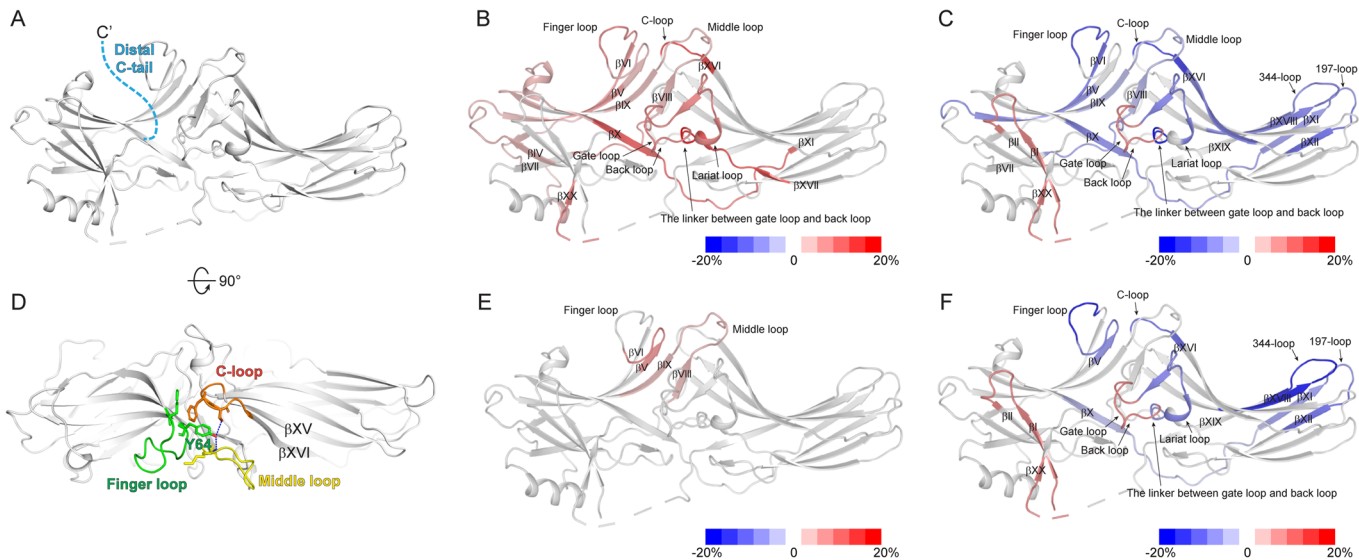

**Figure 2. HDX-MS profile analysis of β-arrestin-2 (βarr2)_1-394 and Y64.**

(**A**) The truncated distal C-tail of βarr2 is colored light blue on the basal state structure of βarr2 (PDB: 3P2D) (Data ref:Zhan et al, 2011a; Zhan et al, 2011b). (**B**) HDX-MS profile comparison between the WT and βarr2_1-394. The HDX-level differences (i.e., HDX levels of WT βarr2–HDX levels of βarr2_1-394) are color-coded on the basal state structure of βarr2 (PDB: 3P2D) (Data ref:Zhan et al, 2011a; Zhan et al, 2011b). Results were derived from three independent experiments (biological). (**C**) HDX-MS profile comparison between apo and PIP$_2$-bound βarr2_1-394. The HDX-level differences (i.e., HDX levels of apo βarr2_1-394 – HDX levels of PIP$_2$-bound βarr2_1-394) are color-coded on the basal state structure of βarr2 (PDB: 3P2D) (Data ref:Zhan et al, 2011a; Zhan et al, 2011b). Results were derived from three independent experiments (biological). (**D**) The top-view of the interaction between the finger, middle, and C-loops of βarr2 in the basal state (PDB: 3P2D) (Data ref:Zhan et al, 2011b). Y64 is indicated by green sticks. The finger, middle, and C-loops are green, yellow, and orange, respectively. (**E**) HDX-MS profiles of the WT and Y64A. HDX-level differences (i.e., HDX levels of WT βarr2–HDX levels of Y64A) are color-coded based on the basal state structure of βarr2 (PDB: 3P2D) (Data ref:Zhan et al, 2011a; Zhan et al, 2011b). Results were derived from three independent experiments (biological). (**F**) HDX-MS profile comparison of apo- and PIP$_2$-bound Y64A. The HDX-level differences (i.e., HDX levels of apo Y64A–HDX levels of PIP$_2$-bound Y64A) are color-coded based on the basal state structure of βarr2 (PDB: 3P2D) (Data ref:Zhan et al, 2011a; Zhan et al, 2011b). Results were derived from three independent experiments (biological). The color-coded HDX-level differences are based on the maximum differences at any D$_2$O incubation time point. The detailed HDX-MS data are summarized in Dataset EV1 and Figs. EV3 and EV4.

conformational changes transmitted from the PIP$_2$-binding sites to the gate loop or βXX.

The initial candidate was the distal C-tail (Fig. 2A). High-resolution structures have not fully characterized the distal C-tail because it is often unresolved or truncated (Han et al, 2001b; Hirsch et al, 1999; Zhan et al, 2011b). Nonetheless, given that the truncation of the distal C-tail transforms βarrs into the pre-active state (Celver et al, 2002; Gurevich, 1998; Gurevich et al, 1997; Kovoor et al, 1999), it is reasonable to hypothesize that the binding of PIP$_2$ perturbs the conformational dynamics of the distal C-tail to impact the activation status of βarrs. To test this hypothesis, we truncated the distal C-tail (βarr2_1-394) and examined PIP$_2$-induced HDX-level changes. If the distal C-tail serves as the route for allosteric conformational changes, PIP$_2$ should not affect HDX levels at the gate loop or βXX in βarr2_1-394.

In the apo state, compared to the wild-type (WT), βarr2_1-394 exhibited higher HDX levels in numerous regions across the N- and C-domains (Figs. 2B and EV3), indicating that the distal C-tail truncation yields βarr2 conformationally more dynamic. This increased conformational dynamics, especially at the gate loop and βXX, accounts for the pre-active state, as previously reported (Celver et al, 2002; Gurevich, 1998; Gurevich et al, 1997; Kovoor et al, 1999).

PIP$_2$ induced HDX-level changes of βarr2_1-394 in the regions similar to the WT (compare Figs. 1E and 2C; Table EV1). Decreased HDX levels were detected at the PIP$_2$-binding site

(Figs. 2C and EV3, peptide 324–338) and increased HDX levels were detected at βI, gate loop, and proximal C-tail through βXX (Fig. 2C and EV3, peptides 1–19, 292–302, and 382–389). These findings suggest that the PIP$_2$ can induce further activation of βarr2_1-394.

Other regions altered in the WT were also similarly affected (Figs. 2C and EV3, peptides 70–76, 75–81, 168–186, 195–201, 246–250, 281–291, and 303–306). A few other regions where we did not observe HDX changes with PIP$_2$-bound WT were also affected (Fig. EV3, peptides 50–64, 118–127, 128–145, and 251–258), but the HDX levels of these regions became statistically no different to those of the WT (Fig. EV3, peptides 128–145 and 251–258) or similar to those of the WT (Fig. EV3, peptides 50–64 and 118–127). In addition, we observed the decreased HDX levels at βXI (Figs. 2C and EV3, peptide 187–194). Overall, the HDX profile changes of the PIP$_2$-bound βarr2_1-394 (Fig. 2C) were similar to those of the PIP$_2$-bound WT (Fig. 1E), suggesting that the distal C-tail is not the route for allosteric conformational changes from the PIP$_2$-binding sites to the gate loop or βXX.

## Y64 in the finger loop is not involved in PIP$_2$-induced pre-activation

The finger, middle, and C-loops between the N- and C-domains undergo dramatic conformational changes upon activation (Fig. 1B) and interact with the cytosolic core of the receptor (Fig. 1C)

(Huang et al, 2020b; Kang et al, 2015). In the basal state, the finger, middle, and C-loops form a designated structure through hydrophobic and polar interactions (Fig. 2D). PIP$_2$ altered HDX levels in the finger loop and C-loop (Fig. 1E). Notably, the C-loop is located at the C-domain as an extension from the PIP$_2$-binding sites (βXV and βXVI) (Figs. 1C and 2D). Therefore, our second hypothesis was that interactions between the finger, middle, and C-loops transmit the allosteric conformational changes. In the basal state, Y64 is located in a pocket formed by the finger-, middle-, and C-loops (Fig. 2D), probably stabilizing the interactions between these three loops. Thus, we reasoned that the mutation of Y64 destabilizes the interactions between these three loops and breaks off the transmission route from the PIP$_2$-binding sites.

In the apo state, the mutation of Y64 to alanine (Y64A) altered HDX levels in the N-terminal half of the finger loop and middle loop compared to those in the WT (Figs. 2E and EV4, peptides 62–69 and 128–145), reflecting a disturbance of the conformation surrounding Y64, as expected. Upon addition of PIP$_2$, Y64A displayed HDX changes in the regions similar to those of the WT (compare Figs. 1E and 2F; Table EV1). HDX levels were decreased at the PIP$_2$-binding site (Figs. 2F and EV4, peptide 324–338) and increased at βI, gate loop, and proximal C-tail through βXX (Figs. 2F and EV4, peptides 1–19, 292–302, and 382–389). These findings suggest that PIP$_2$ can induce pre-activation of Y64A. Other

regions altered in the WT were also affected (Figs. 2F and EV4, peptides 62–69, 70–76, 168–186, 195–201, 246–250, 281–291, and 303–306). In addition, we observed decreased HDX levels at βXI (Fig. 2F and EV4, peptides 187–194). These results suggest that the interactions between the finger, middle, and C-loops are not routes for allosteric conformational transmission.

## The lariat loop of βarr2 is involved in PIP$_2$-induced pre-activation

Because the distal C-tail and the interactions between the finger, middle, and C-loops do not serve as routes for allosteric conformational transmission, we sought other potential routes. After careful examination of the basal state structure and HDX-MS data of PIP$_2$-bound βarr2 (Figs. 1E and EV2), L280 in the lariat loop and E315 in the back loop were chosen as potential key residues (Fig. 3).

In the basal state, L280 faces the gate loop and forms hydrophobic interactions with L291, L295, and L302 (Fig. 3A), which stabilizes the conformation of the gate and lariat loops. If the allosteric conformational transmission is mediated through perturbation of the interaction of the lariat and gate loops, the L280 mutation would disrupt this route. To test this hypothesis, we mutated L280 to glycine.

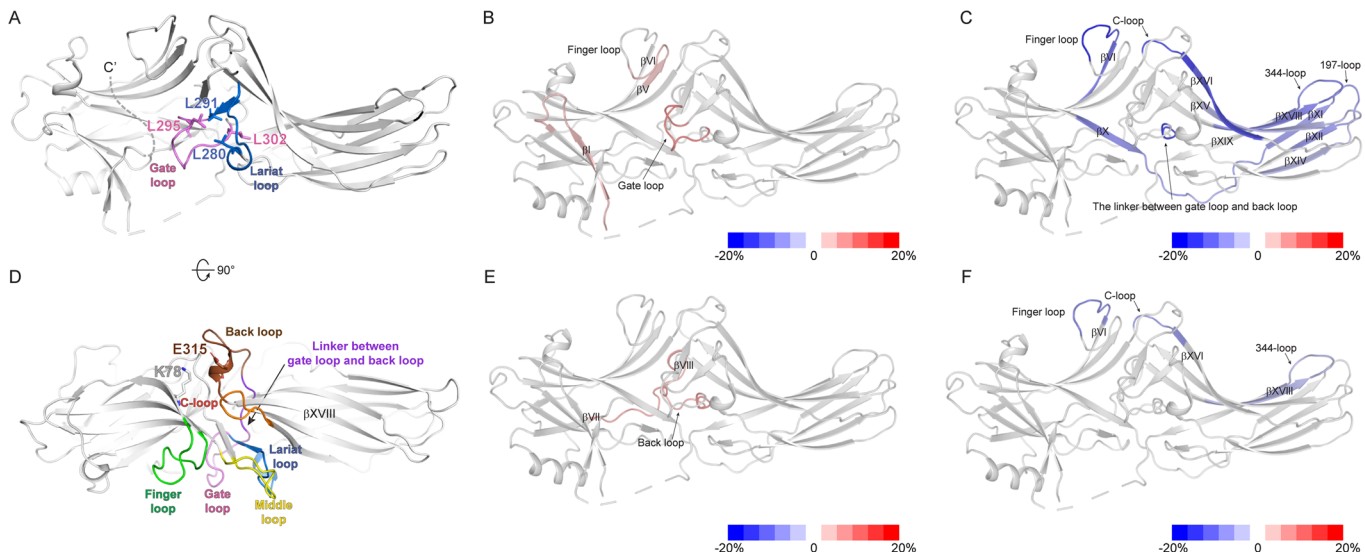

Figure 3. HDX-MS profile analysis of L280 and E315.

(A) Interaction between the lariat and gate loops in the basal state βarr2 (PDB: 3P2D) (Data ref:Zhan et al, 2011a; Zhan et al, 2011b). The lariat and gate loops are shown in blue and pink, respectively. The hydrophobic residues forming the interaction between the lariat and gate loops are shown as sticks. (B) HDX-MS profile comparison of the WT and L280G. The HDX-level differences (i.e., HDX levels of WT βarr2–HDX levels of L280G) are color-coded on the basal state structure of βarr2 (PDB: 3P2D) (Data ref:Zhan et al, 2011a; Zhan et al, 2011b). Results were derived from three independent experiments (biological). (C) HDX-MS profile comparison of apo and PIP$_2$-bound L280G. The HDX-level differences (i.e., HDX levels of apo Y64A–HDX levels of PIP$_2$-bound L280G) are color-coded on the basal state structure of βarr2 (PDB: 3P2D) (Data ref:Zhan et al, 2011a; Zhan et al, 2011b). Results were derived from three independent experiments (biological). (D) Top-view of the structure of the basal state βarr2 (PDB: 3P2D) (Data ref:Zhan et al, 2011a; Zhan et al, 2011b) showing relative positions of the back (brown), gate (pink), and lariat (blue) loops, and the linker between the gate and back loops (violet). E315 is shown as brown sticks, and K78 is shown as gray sticks. The finger, middle, and C-loops are colored green, yellow, and orange, respectively. (E) HDX-MS profile comparison between the WT and E315A. The HDX-level differences (i.e., HDX levels of WT βarr2–HDX levels of E315A) are color-coded on the basal state structure of βarr2 (PDB: 3P2D) (Data ref:Zhan et al, 2011a; Zhan et al, 2011b). Results were derived from three independent experiments (biological). (F) HDX-MS profile comparison of apo and PIP$_2$-bound E315A. The HDX-level differences (i.e., HDX levels of apo Y64A–HDX levels of PIP$_2$-bound E315A) are color-coded on the basal state structure of βarr2 (PDB: 3P2D) (Data ref:Zhan et al, 2011a; Zhan et al, 2011b). Results were derived from three independent experiments (biological). The color-coded HDX-level differences are based on the maximum differences at any D$_2$O incubation time point. The detailed HDX-MS data are summarized in Dataset EV1 and Figs. EV4 and EV5.

In the apo state, L280G showed altered HDX levels at the gate loop (Figs. 3B and EV4, peptide 292–302), reflecting the altered conformation near the lariat and gate loop regions due to the mutation. L280G mutation also altered HDX levels at βI and the N-terminal half of the finger loop (Figs. 3B and EV4, peptides 1–19 and 62–69). The results suggest that perturbation of the interaction between the gate and the lariat loops could alter the conformational dynamics of remote regions, such as βI and finger loop.

Upon PIP$_2$-binding to L280G, we observed decreased HDX levels at the PIP$_2$-binding interface (Figs. 3C and EV4, peptide 324–338). We also observed altered HDX at the regions similar to those of the WT (compare Figs. 1E and 3C; Table EV1), such as the C-terminal half of the finger loop, βVI, βX through βXI, 197-loop, C-loop, and the linker between lariat loop and back loop (Figs. 3C and EV4, peptides 70–76, 75–81, 168–186, 195–201, 246–250, and 303–306). In addition, decreased HDX levels were evident at βXI, βXIV through βXV, and βXVI (Figs. 3C and EV4, peptides 187–194, 219–239, and 251–258).

In contrast, changes in HDX levels for βI, the gate loop, and proximal C-tail through βXX were not evident upon the binding of PIP$_2$ to L280G (Figs. 3C and EV4, peptides 1–19, 292–302, and 382–389). Therefore, the binding of PIP$_2$ in L280G induces conformational changes in most regions similar to those of the WT but failed to transform it to the pre-active conformation (i.e., disturbance of the gate loop and βXX), suggesting that perturbation of the interaction of the lariat and gate loops is the route for the transmission of the conformational changes from the PIP$_2$-binding site to βXX.

## The back loop of βarr2 is involved in PIP$_2$-induced pre-activation

Another potential route we examined was the back loop. Although the HDX-MS profiles of the back loop were not affected by PIP$_2$, the neighboring C-loop and the linker between the lariat and back loops were altered (Figs. 1E and EV2, peptides 246–250 and 303–306). Interestingly, the back loop is an extension of the PIP$_2$-binding sites (βXVIII), located adjacent to the C-loop, and directly connected to the gate loop through the linker between the gate loop and the back loop (Fig. 3D). Previous evidence suggested that in the basal state, E315 at the back loop occasionally forms salt bridge with K78 at βVI (Fig. 3D) and that disruption of this interaction results in ligand-independent accumulation of βarr2 in the clathrin-coated endocytic structures (Eichel et al, 2018). Moreover, the back loop has been reported as a potential route for the conformational transition from PIP$_2$-binding to βarr1 C-tail release (Zhai et al, 2023). Therefore, we further examined the role of the back loop in the PIP$_2$-induce βarr2 activation.

To test this hypothesis, we mutated E315 to alanine, which would break the interaction between E315 and Y78 (Fig. 3D). In the apo state, compared to the WT, E315A showed higher HDX levels at the back loop and its neighboring βVII/βVIII loop (Figs. 3E and EV5, peptides 118–127 and 303–317) reflecting altered conformational dynamics of the back loop upon E315A mutation. As we did not observe HDX differences in other regions remote from the back loop, the results imply that the disruption of the interaction between E315 and K78 alters the local conformational dynamics but does not affect the overall conformational dynamics of βarr2.

Although the apo state did not show HDX-level differences between the WT and E315A other than in the back loop and βVII/

βVIII loop, the effects of PIP$_2$ on E315A were dramatically different from those on the WT (compare Figs. 1E and 3F; Table EV1). Although PIP$_2$ induced a decrease in HDX levels at the PIP$_2$-binding site in E315A (Figs. 3F and EV5, peptide 324–338), indicating PIP$_2$-binding to E315A, we observed HDX-MS profile changes only within very limited regions, such as the C-terminal half of the finger loop and the C-loop (Figs. 3F and EV5, peptides 70–76 and 246–250) but no other regions. These results suggest that, in E315A, PIP$_2$ could alter the C-loop and its neighboring finger loop but fails to transform βarr2 to the pre-active state. Thus, we conclude that the PIP$_2$-induced conformational changes are allosterically transmitted through the back loop to βXX.

## PIP$_2$ facilitates V2Rpp-induced βarr2 activation

A recent study by Zhai et al reported that the simultaneous binding of V2Rpp and PIP$_2$ induces complex conformational changes in different structural regions (Zhai et al, 2023). Here, we tested whether pre-incubation with PIP$_2$ affects the V2Rpp-induced C-tail release. To examine the C-tail release of βarr2, we developed an experimental system using bimane fluorophore, an environment-sensitive fluorescent molecule. We substituted glycine at residue 6 to tryptophan (G6W) and labeled bimane at the βarr2 C-tail by substituting aspartate at residue 386 with cysteine (D386C) (Fig. 4A) in the cysteine-free βarr2 background (Cys-free βarr2: C17S/C60A/C126S/C141I/C151V/C244V/C253V/C271S/C405S/C410S). Bimane fluorescence can be quenched by nearby tryptophan residues (Jones Brunette and Farrens, 2014). Therefore, in the basal state, bimane fluorescence at the residue 386 is quenched by the tryptophan at the residue 6 (Fig. 4A, upper panel), but upon C-tail release, quenching is abolished as the residue 386 moves away from the residue 6 (Fig. 4A, lower panel).

When we incubated the bimane-labeled βarr2 with excess V2Rpp (300 μM), bimane fluorescence increased (Fig. 4B), reflecting the C-tail release. To examine the pre-activation effect of PIP$_2$, we reduced V2Rpp concentration to 30 μM, where it induces minimal bimane fluorescence increase (Fig. 4C). Similarly, 30 μM PIP$_2$ did not induce C-tail release (Fig. 4C). However, pre-incubation with PIP$_2$ followed by V2Rpp addition significantly increased the bimane fluorescence (Fig. 4C), implying that PIP$_2$ pre-incubation facilitates V2Rpp-induced βarr2 activation.

Introducing the L280G mutation to disrupt the allosteric conformational pathway reduced the augmentation of the V2Rpp-induced C-tail release after PIP$_2$ pre-incubation (Fig. 4C). This result confirms that the lariat loop is the allosteric conformational change route through which PIP$_2$ facilitates phosphorylated receptor-induced βarr2 activation.

## Proposed mechanism of the PIP$_2$-induced βarr2 activation

Here, we comprehensively analyzed the conformational dynamics of the whole βarr2 using HDX-MS. Our data confirmed that PIP$_2$-bound βarr2 adopts the pre-active conformation, enhancing βarr2's interaction with V2Rpp. Interestingly, the PIP$_2$-binding sites are remote from βXX and polar core (Fig. 4D), suggesting allosteric conformational transmission. The HDX-MS data suggested that both L280G and E315A failed to activate βarr2, but the

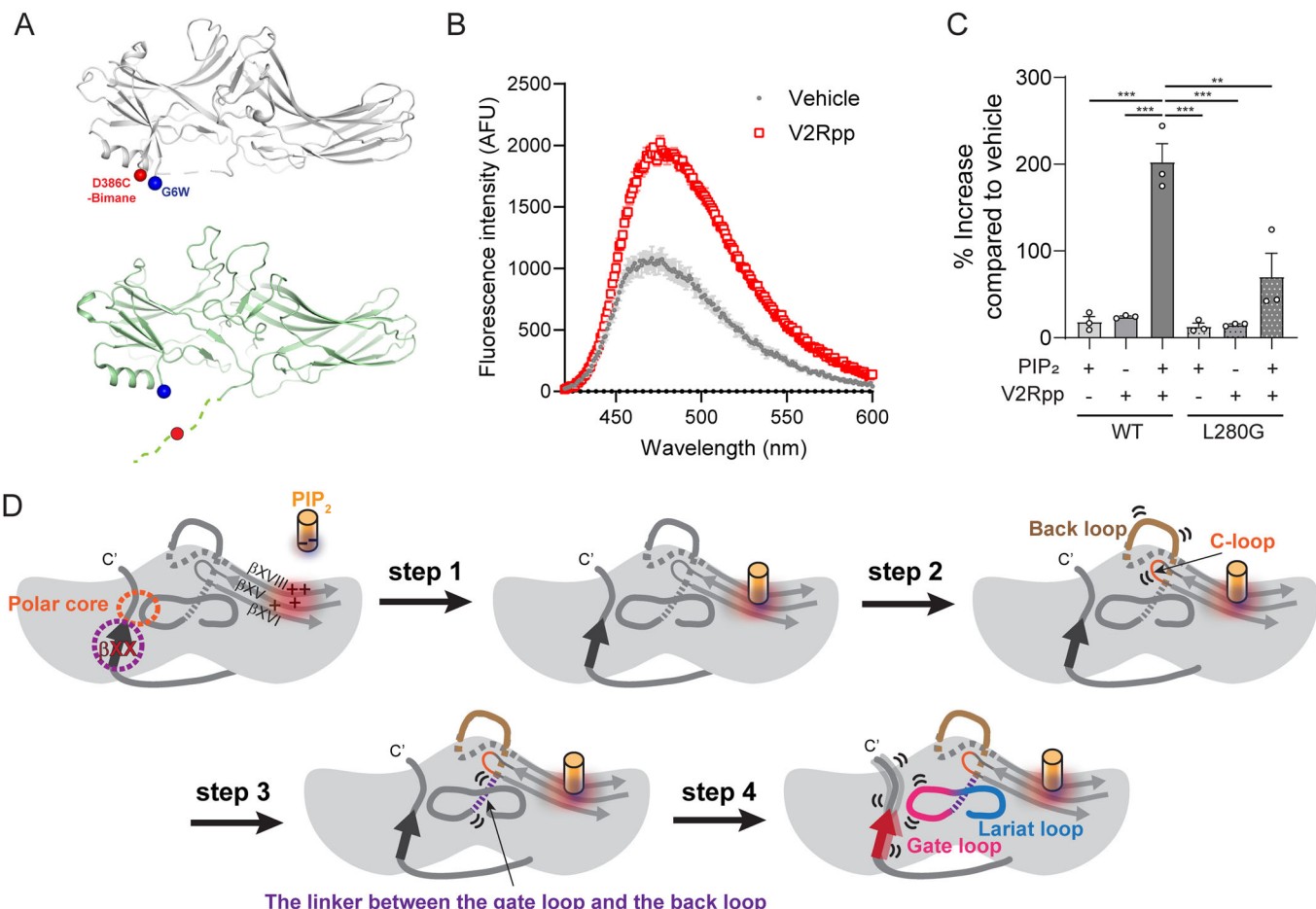

**Figure 4. Proposed molecular mechanism of βarr2 pre-activation upon PIP$_2$-binding.**

(A) The scheme of the experimental system analyzing C-tail release. The position of residues 6 and 386 are shown as red and blue spheres in the basal state structure of βarr2 (upper panel, PDB: 3P2D) (Data ref:Zhan et al, 2011a; Zhan et al, 2011b) and V2Rpp-bound structure of βarr2 (lower panel, PDB: 8GOC) (Data ref:Maharana et al, 2023a; Maharana et al, 2023b). (B) Bimane fluorescence traces of basal and V2Rpp-bound states. (C) Bimane fluorescence changes of wild-type (WT) or L280G βarr2 upon addition of PIP$_2$, V2Rpp, or PIP$_2$ pre-incubation followed by V2Rpp addition. The statistical significance of the differences was determined using one-way ANOVA followed by Tukey's posttest (**$P < 0.001$ and ***$P < 0.0001$). Exact $P$-values between PIP$_2$-WT vs. PIP$_2$-V2Rpp-WT, V2Rpp-WT vs. PIP$_2$-V2Rpp-WT, PIP$_2$-V2Rpp-WT vs. PIP$_2$-L280G, PIP$_2$-V2Rpp-WT vs. V2Rpp-L280G, and PIP$_2$-V2Rpp-WT vs. PIP$_2$-V2Rpp-L280G are 0.000017, 0.000016, 0.000009, 0.000010, and 0.000317, respectively. Results were derived from three independent experiments (biological). Data are presented as mean ± standard error of the mean. (D) βArr2 is shown schematically as a gray shape, particular regions are highlighted in dark gray (dotted) lines. Secondary structures (beta sheets) of interest are shown as arrows. The positively charged region in C-domain is indicated by red color and "+" symbols. PIP$_2$ is shown as the orange cylinder. In its basal state, βarr2 is stabilized through the interaction of βXX at the N-domain (dotted purple circle) and through the ionic interactions between N- and C-domains (i.e., polar core, dotted orange circle). The negatively charged PIP$_2$ binds to the positively charged region within the C-domain of βarr2 (step 1). This PIP$_2$-binding induces alterations in the conformational dynamics of the back and C-loops (step 2). The resulting conformational change is transmitted to the linker between the gate loop and the back loop (step 3). Consequently, the conformational dynamics of the gate and lariat loops are altered, leading to the destabilization of the gate loop and βXX, ultimately resulting in the pre-activation of βarr2 (step 4).

two mutants exhibited different PIP$_2$-induced conformational changes (Fig. 3). In L280G, PIP$_2$ could still induce conformational changes in almost all the regions similar to the WT, except the gate loop, βXX, and βI (Fig. 3C). In contrast, in E315A, the binding of PIP$_2$ induced conformational changes in only limited regions (i.e., the C-loop and its neighboring finger loop; Fig. 3F) without affecting other regions. Thus, we propose that the back loop precede the gate loop when the allosteric conformational changes are transmitted from the PIP$_2$-binding site to βXX. In summary, the binding of PIP$_2$ at the C-domain (Fig. 4D, step 1) affects the loops (i.e., the back loop and C-loop) that are connected to the PIP$_2$-binding β-strands (Fig. 4D, step 2). The altered

conformational dynamics of the back loop is allosterically transmitted to the lariat and gate loops to pre-activate βarr2 (Fig. 4D, step 4) through the linker between the gate and back loops (Fig. 4D, step 3).

## Limitations and future perspectives

This study highlights the structural mechanism of the PIP$_2$-induced βarr2 pre-activation but has limitations. First, within the cell, arrestins interact with a variety of other components, including phospholipids, receptors, G proteins, and signaling proteins (Chen et al, 2023b; Grimes et al, 2023; Lally et al, 2017; Qu et al, 2021b;

Smith et al, 2021). Therefore, the PIP$_2$- or V2Rpp-induced conformational changes of the purified βarr2 might be too simplistic compared to the complex nature within the cell. Second, our study couldn't detail the allosteric conformation changes at the atomic level. Advances in biophysical techniques, such as time-resolved Cryo-EM (Klebl et al, 2023), could provide deeper insights into the step-by-step conformational changes at the atomic level.

It has long been believed that the interaction of the phosphorylated GPCRs at the N-domain is the key process for arrestin activation (Edward Zhou et al, 2019; Gusach et al, 2023; Hilger et al, 2018; Maharana et al, 2022; Seyedabadi et al, 2021; Wisler et al, 2014; Zhao et al, 2017). However, now it is emerging that arrestin activation can be achieved through various processes. Inositol hexaphosphate (IP$_6$) interacts at the phosphate sensor within the N-domain resulting βXX release to activate βarr2 and triggers further downstream signal transduction (Chen et al, 2017). PIP$_2$ has been suggested to interact at the C-domain to activate arrestins (Janetzko et al, 2022; Zhai et al, 2023), and here we further propose the structural mechanism of PIP$_2$-induced arrestin pre-activation. As it is evident that arrestins can be activated via various routes, it is needed to investigate the structural differences and the functional consequences of the different active states of arrestins.

# Methods

## βarr2 expression and purification

All protein constructs for HDX-MS were cloned into the pET28a vector, and mutant rat βarr2 constructs for Trp-induced bimane fluorescence quenching experiments were cloned into the pET28b. The rat βarr2 constructs were transformed into *Escherichia coli* BL21 (DE3). Point mutations were prepared using site-directed mutagenesis. Expression and purification were performed as previously described (Park et al, 2019). Briefly, WT rat βarr2 and the mutants were grown in LB broth medium at 37 °C until the optical density at 600 nm reached 0.4–0.6. The bacteria were then induced with 30 µM IPTG for 24 h at 16 °C. Proteins were purified using Ni-IDA resins and size-exclusion chromatography.

## Hydrogen/deuterium exchange

βarr2 at a final concentration of 50 µM in 20 mM HEPES pH 7.4, 150 mM NaCl, and 100 µM Tris [2-carboxyethyl] phosphine hydrochloride was co-incubated with 500 µM V2Rpp or 150 µM PIP$_2$ for 1 h at room temperature. HDX was performed by mixing 2 µL of protein (50 µM) with 28 µL of D$_2$O buffer (20 mM HEPES pH 7.4, 150 mM NaCl, 100 µM Tris [2-carboxyethyl] phosphine hydrochloride, and 10% glycerol in D$_2$O) and incubating the mixture for 10, 100, 1000, and 10,000 s on ice. At the indicated time points, the reaction was quenched by adding 30 µL of ice-cold quench buffer (60 mM NaH$_2$PO$_4$ pH 2.01 and 10% glycerol) and snap-frozen on dry ice. Identical procedures were conducted for

### Reagents and tools table

| Reagent/resource | Reference or source | Identifier or catalog number |
|---|---|---|
| **Experimental models** | | |
| *E. coli* DH5α Chemically Competent | Enzynomics | Cat# CP010 |
| *E. coli* ROSETTA(DE3) | Novagen | Cat# 70954 |
| **Recombinant DNA** | | |
| pET28a-WT rat β-arrestin-2 | This study | N/A |
| pET28a-β-arrestin 2_1-394 | This study | N/A |
| pET28a-β-arrestin 2_Y64A | This study | N/A |
| pET28a-β-arrestin 2_L280G | This study | N/A |
| pET28a-β-arrestin 2_E315A | This study | N/A |
| pET28b-rat β-arrestin-2 _Cysfree_G6W_D386C | This study | N/A |
| pET28b-rat β-arrestin-2 _Cysfree_G6W_L280G_D386C | This study | N/A |
| **Oligonucleotides and other sequence-based reagents** | | |
| PCR primer: Y64A Forward: GTGCCTTCCGCGCTGGCCGAGAAGACCTGGATG | Bionics | N/A |
| PCR primer: L280G Forward: CACCATAACCCCGCTGGGCAGTGACAACCGAGAGAAG | Bionics | N/A |
| PCR primer: E315A Forward: GAGGGAGCCAACAAGGCGGTGCTGGGAATCCTAG | Bionics | N/A |
| PCR primer: G6W Forward: CATATGGGTGAGAAGCCCTGGACCAGGGTCTTCAAG | Bionics | N/A |
| PCR primer: D386C Forward: CCAACTATGCCACAGACGACTGCATCGTGTTTGAGGAC | Bionics | N/A |
| **Chemicals, enzymes and other reagents** | | |
| Protease inhibitor cocktail | BioVision | Cat# K272 |
| Deuterium oxide | Cambridge isotope laboratories | Cat# DLM-11-100 |
| Ni-IDA resin | Cytiva | Cat# 17057501 |

| Reagent/resource | Reference or source | Identifier or catalog number |
|---|---|---|
| Dimethyl sulfoxide | Duchefa | Cat# D1370 |
| Leupeptin | Goldbio | Cat# L-010 |
| Lysozyme | Goldbio | Cat# L-040 |
| TCEP | Goldbio | Cat# TCEP |
| IPTG | Goldbio | Cat# I2481C |
| Kanamycin | Goldbio | Cat# K-120-10N |
| Chloramphenicol | Duchefa | Cat# C0113.0025 |
| Pepsin column | Life Technologies | Cat# 2313100 |
| DNase I | Roche | Cat# 11284932001 |
| Benzamidine | Sigma-Aldrich | Cat# 12072 |
| Imidazole | Sigma-Aldrich | Cat# I2399 |
| 08:0 $PI(4,5)P_2$ | Avanti Polar Lipids | Cat# 850185 |
| Zeba Desalt Spin Desalting Columns | Thermo Scientific | Cat# 89890 |
| Bromobimane | MedChemExpress | Cat# HY-100041 |
| V2Rpp | Tufts University Core Facility | N/A |
| **Software** | | |
| Prism 8.0 | Graphpad | graphpad.com |
| PyMol 2.3 | Schrodinger | pymol.org |
| Proteinlynx Global Server 2.4 | Waters | www.waters.com |
| DynamX 3.0 | Waters | www.waters.com |

nondeuterated samples using a $H_2O$ buffer comprising 20 mM HEPES, pH 7.4, 150 mM NaCl, and 10% glycerol in $H_2O$.

## HDX-MS

HDX-MS and data analyses were conducted as previously described (Du et al, 2019; Qu et al, 2021a). Briefly, the quenched samples underwent online digestion by passage through an immobilized pepsin column (2.1 × 30 mm; Life Technologies, Carlsbad, CA, USA). The digested peptide fragments were collected on a C18 VanGuard trap column (1.7 mm × 30 mm; Waters, Milford, MA, USA), followed by ultra-pressure liquid chromatography using an ACQUITY UPLC C18 column (1.7 mm, 1.0 mm × 100 mm; Waters). All settings and conditions for the system, such as voltage, temperature, collision energy, and lockspray, were as previously reported (Du et al, 2019; Qu et al, 2021a). Peptic peptides from nondeuterated samples were identified using ProteinLynx Global Server 2.4 (Waters). To process HDX-MS data, the amount of deuterium in each peptide was determined by measuring the centroid of the isotopic distribution using DynamX 3.0 (Waters).

## Trp-induced bimane fluorescence quenching experiment

βarr2 was prepared at a final concentration of 8 μM or 20 μM in 20 mM HEPES pH 7.4, 150 mM NaCl, and co-incubated with a 10-fold molar excess of bromobimane for 1 h on ice. Excess dye was removed by buffer exchange using a desalting column. Then, βarr2 at a final concentration of 3 μM was co-incubated with V2Rpp for 1 h at room temperature, with or without pre-incubation of $PIP_2$. The samples were placed in a MicroFluor 96-well fluorescent plate. The samples were excited at 390 nm, and the emitted fluorescence was measured from 420 to 600 nm using 1-nm step size by Synergy Neo or Synergy Neo2 (BioTek, Winooski, VT, USA).

## Statistical analysis

For HDX-MS analysis, mass differences >0.22 Da and 2% were considered significant. Student's $t$ test was used to determine the statistically significant differences between individual time points. For Trp-induced bimane fluorescence quenching data, the significant differences were analyzed by one-way ANOVA followed by Tukey's posttest. The statistical analyses were performed by GraphPad Prism software (GraphPad, San Diego, CA, USA), and statistical significance was set at $P < 0.05$.

## Data availability

HDX-MS data have been deposited to ProteomeXchange Consortium via PRIDE43 partner repository with the set identifier PXD049391 (https://www.ebi.ac.uk/pride/archive/projects/PXD049391).

The source data of this paper are collected in the following database record: biostudies:S-SCDT-10_1038-S44319-024-00239-x.

# Peer review information

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

## Acknowledgements

This work was supported by grants from the National Research Foundation of Korea, funded by the Korean government (NRF-2021R1A2C3003518 and NRF-2019R1A5A2027340 to KYC) and by a grant from the Ministry of Oceans and Fisheries' R&D project, Korea (2021633 to KYC).

## Author contributions

**Kiae Kim**: Data curation; Formal analysis; Writing—original draft. **Ka Young Chung**: Conceptualization; Data curation; Formal analysis; Supervision; Funding acquisition; Investigation; Writing—original draft; Project administration.

Source data underlying figure panels in this paper may have individual authorship assigned. Where available, figure panel/source data authorship is listed in the following database record: biostudies:S-SCDT-10_1038-S44319-024-00239-x.

## Disclosure and competing interests statement

Kiae Kim and Ka Young Chung has a patent pending for βarr C-tail release assay.

# Expanded View Figures

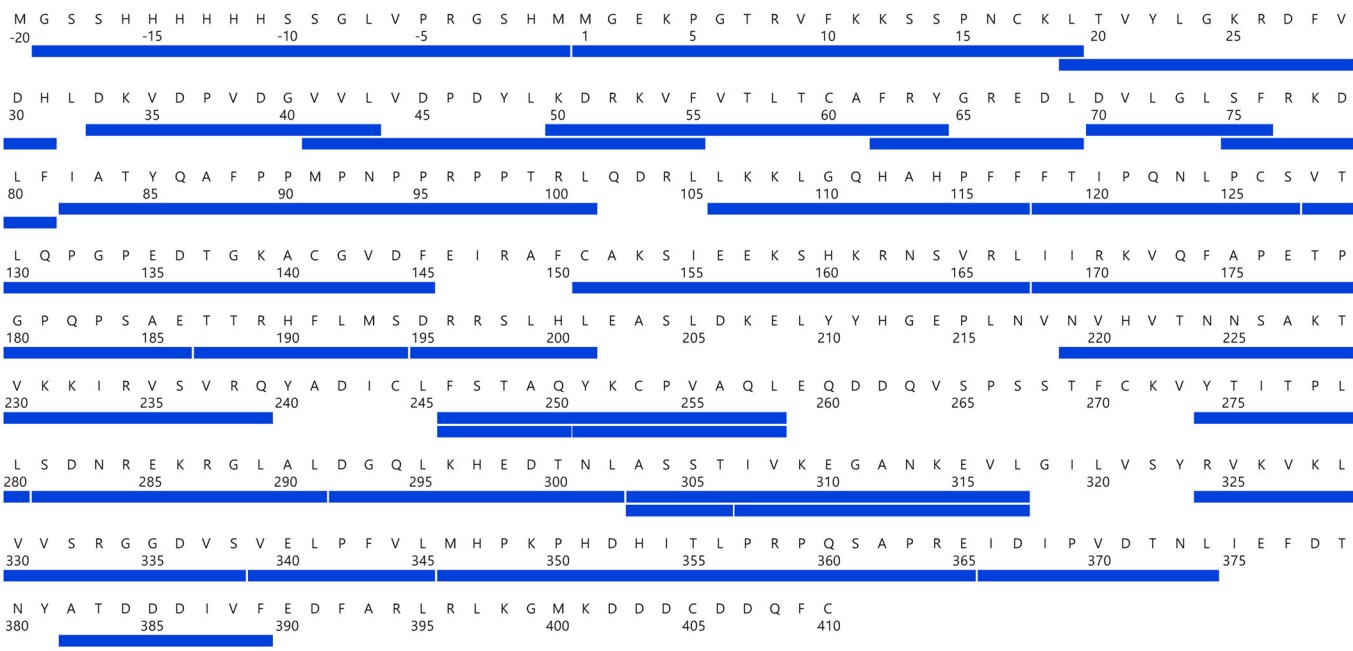

**Figure EV1. Sequence coverage map of wild-type β-arrestin-2 (βarr2).**

The blue bars indicate analyzed peptic peptides.

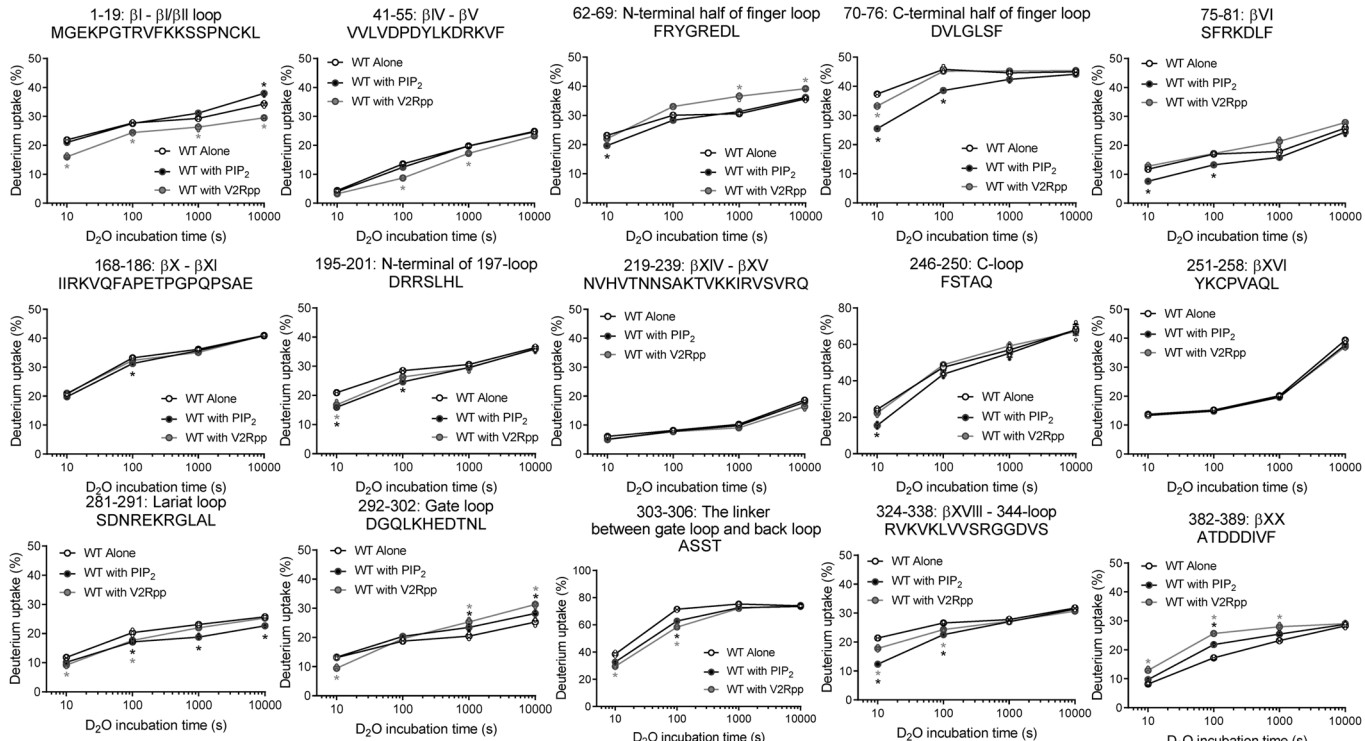

**Figure EV2. Deuterium uptake plots of selective peptides of WT βarr2 with or without V2Rpp or PIP₂ co-incubation.**

Results were derived from three independent experiments. The statistical significance of the differences was determined using Student's t test (*P < 0.05). Exact p-values are provided in Dataset EV1. Data are presented as mean ± standard error of the mean. Black or grey * indicates statistically significant difference between apo WT βarr2 and PIP₂-bound WT βarr2 or V2Rpp-bound WT βarr2, respectively. Smaller symbols indicate each data point.

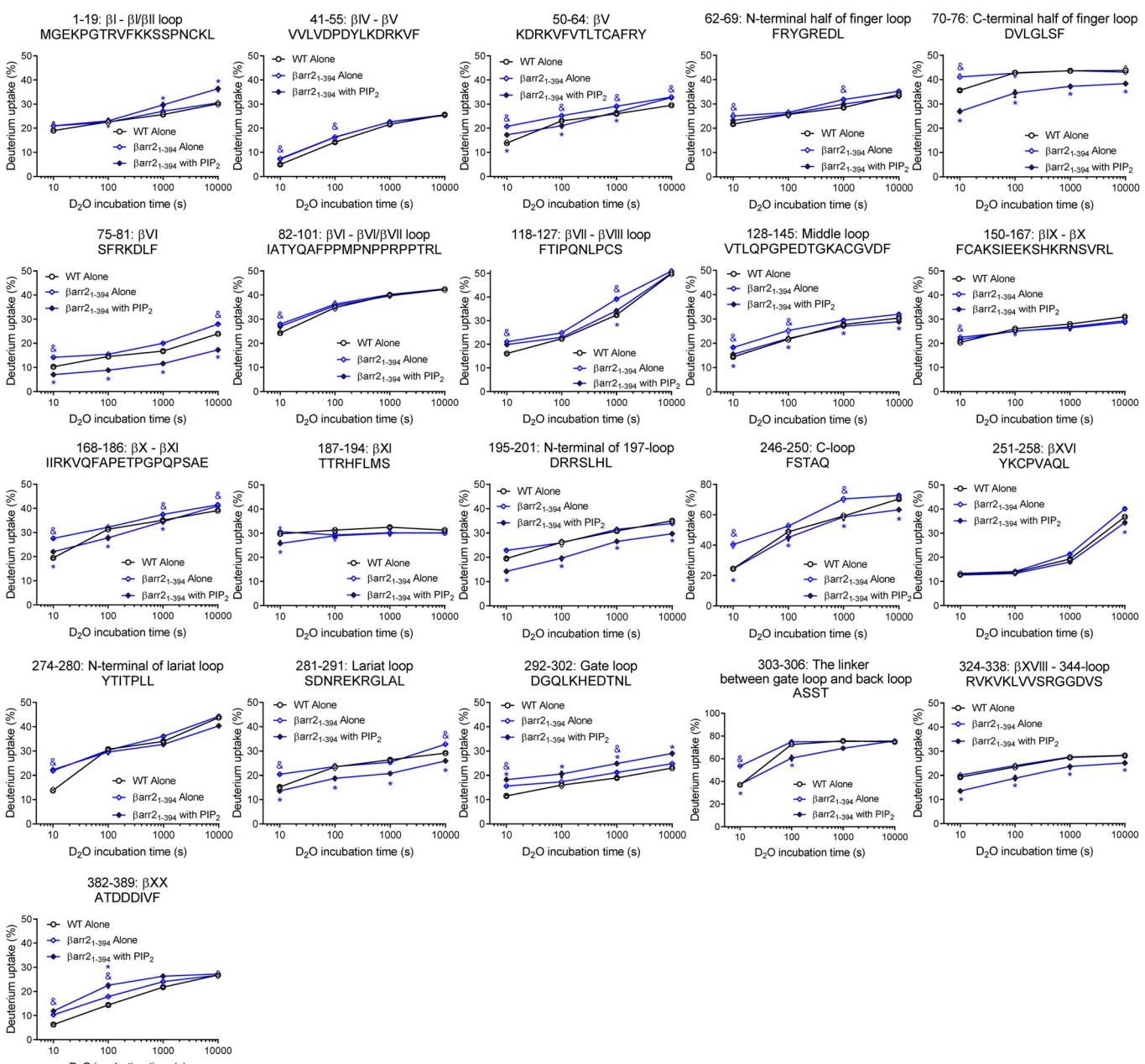

**Figure EV3. Deuterium uptake plots of selective peptides of apo WT βarr2 and βarr2_1-394 with or without PIP₂ co-incubation.**

Results were derived from three independent experiments. The statistical significance of the differences was determined using Student's *t* test ($^{&,*}P < 0.05$). Exact *p*-values are provided in Dataset EV1. & indicates statistically significant difference between apo WT βarr2 and apo βarr2_1-394. * indicates statistically significant difference between apo and PIP₂-bound βarr2_1-394. Data are presented as mean ± standard error of the mean. Smaller symbols indicate each data point.

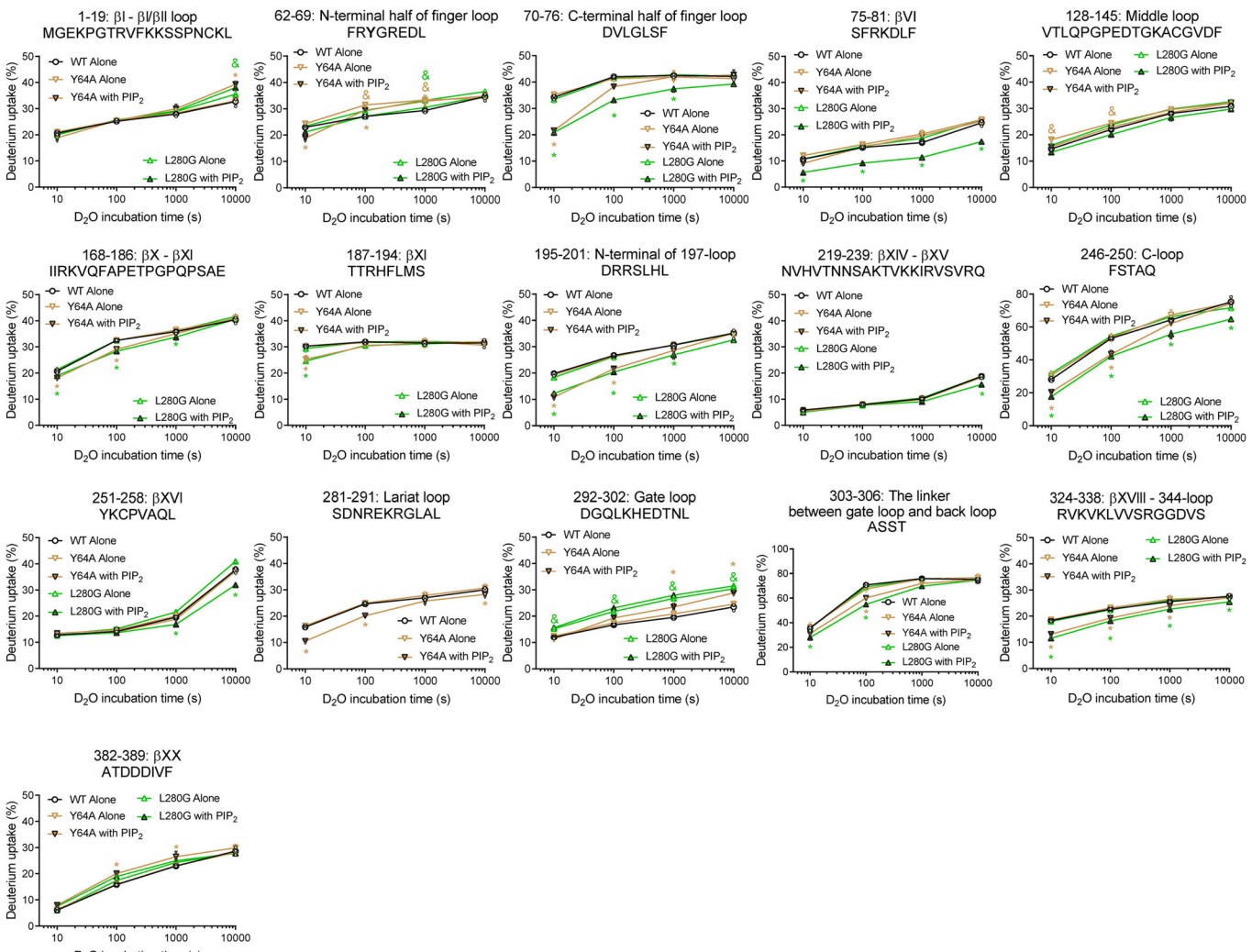

**Figure EV4. Deuterium uptake plots of selective peptides of apo WT βarr2 and Y64A and L280G with or without PIP₂ co-incubation.**

Results were derived from three independent experiments. The statistical significance of the differences was determined using Student's *t* test ($^{&,*}P < 0.05$). Exact *p*-values are provided in Dataset EV1. Yellow green & indicates statistically significant difference between apo WT βarr2 and apo Y64A or L280G, respectively. Yellow or green * indicates statistically significant difference between apo and PIP₂-bound Y64A or L280G, respectively. Data are presented as mean ± standard error of the mean. Smaller symbols indicate each data point.

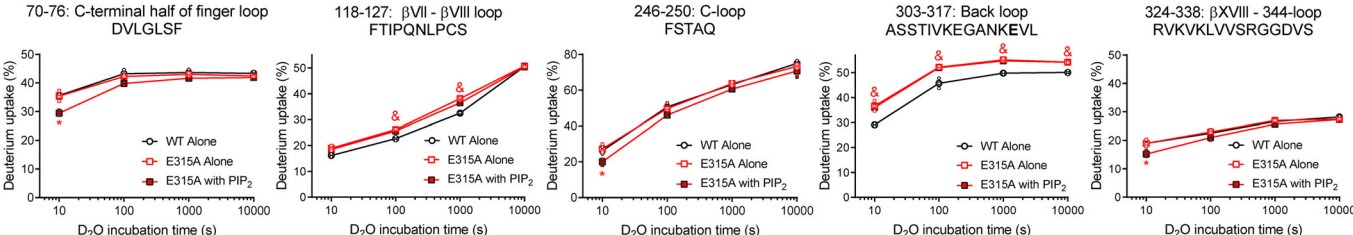

**Figure EV5. Deuterium uptake plots of selective peptides of apo WT βarr2 and E315A with or without PIP₂ co-incubation.**

Results were derived from three independent experiments. The statistical significance of the differences was determined using Student's *t* test ($^{\&,*}P < 0.05$). Exact *p*-values are provided in Dataset EV1. & indicates statistically significant difference between apo WT βarr2 and apo E315A. * indicates statistically significant difference between apo and PIP₂-bound E315A. Data are presented as mean ± standard error of the mean. Smaller symbols indicate each data point.

