## [Peer Review File · EMBO Reports]

Molecular mechanism of β -arrestin-2 pre-activation by phosphatidylinositol 4,5-bisphosphate

Kiae Kim and Ka Young Chung

Corresponding author(s): Ka Young Chung (kychung2@skku.edu)

Review Timeline:

Submission Date:	12th Dec 23
Editorial Decision:	6th Feb 24
Revision Received:	20th May 24
Editorial Decision:	2nd Jul 24
Revision Received:	9th Jul 24
Accepted:	9th Aug 24

Editors: *Martina Rembold and Deniz Senyilmaz Tiebe*

Transaction Report:

Dear Prof. Chung

Thank you for the submission of your research manuscript to our journal. We have now received the full set of referee reports that is copied below.

As you will see, the referees acknowledge that the findings are potentially interesting, but they also raise a number of concerns and have several suggestions, which all need to be addressed. Please also add experimental validation/testing of the proposed model, as suggested by referee #2.

Given these constructive comments, we would like to invite you to revise your manuscript with the understanding that the referee concerns (as detailed above and in their reports) must be fully addressed and their suggestions taken on board. Please address all referee concerns in a complete point-by-point response. Acceptance of the manuscript will depend on a positive outcome of a second round of review. It is EMBO Reports policy to allow a single round of revision only and acceptance or rejection of the manuscript will therefore depend on the completeness of your responses included in the next, final version of the manuscript.

We realize that it is difficult to revise to a specific deadline. In the interest of protecting the conceptual advance provided by the work, we recommend a revision within 3 months (May 6th). Please discuss the revision progress ahead of this time with the editor if you require more time to complete the revisions.

I am also happy to discuss the revision further via e-mail or a video call, if you wish.

*****IMPORTANT NOTE:

We perform an initial quality control of all revised manuscripts before re-review. Your manuscript will FAIL this control and the handling will be delayed IN CASE the following APPLIES:

- 1) A data availability section providing access to data deposited in public databases is missing. If you have not deposited any data, please add a sentence to the data availability section that explains that.
- 2) Your manuscript contains statistics and error bars based on $n=2$. Please use scatter blots in these cases. No statistics should be calculated if $n=2$.

When submitting your revised manuscript, please carefully review the instructions that follow below. Failure to include requested items will delay the evaluation of your revision.*****

- 1) a .docx formatted version of the manuscript text (including legends for main figures, EV figures and tables). Please make sure that the changes are highlighted to be clearly visible.
- 2) individual production quality figure files as .eps, .tif, .jpg (one file per figure). Please download our Figure Preparation Guidelines (figure preparation pdf) from our Author Guidelines pages <https://www.embopress.org/page/journal/14693178/authorguide> for more info on how to prepare your figures.
- 3) a .docx formatted letter INCLUDING the reviewers' reports and your detailed point-by-point responses to their comments. As part of the EMBO Press transparent editorial process, the point-by-point response is part of the Review Process File (RPF), which will be published alongside your paper.
- 4) a complete author checklist, which you can download from our author guidelines (). Please insert information in the checklist that is also reflected in the manuscript. The completed author checklist will also be part of the RPF.
- 5) Please note that all corresponding authors are required to supply an ORCID ID for their name upon submission of a revised manuscript (). Please find instructions on how to link your ORCID ID to your account in our manuscript tracking system in our

Author guidelines

()

6) We replaced Supplementary Information with Expanded View (EV) Figures and Tables that are collapsible/expandable online. A maximum of 5 EV Figures can be typeset. EV Figures should be cited as 'Figure EV1, Figure EV2' etc... in the text and their respective legends should be included in the main text after the legends of regular figures.

7) Please note that a Data Availability section at the end of Materials and Methods is now mandatory. In case you have no data that requires deposition in a public database, please state so instead of refereeing to the database.

See also < <https://www.embopress.org/page/journal/14693178/authorguide#dataavailability>>. Please note that the Data Availability Section is restricted to new primary data that are part of this study.

Additional information on source data and instruction on how to label the files are available .

10) Figure legends and data quantification:

- the name of the statistical test used to generate error bars and P values,
- the number (n) of independent experiments (please specify technical or biological replicates) underlying each data point,
- the nature of the bars and error bars (s.d., s.e.m.)

- If the data are obtained from n {less than or equal to} 5, show the individual data points in addition to the SD or SEM.

- If the data are obtained from n {less than or equal to} 2, use scatter blots showing the individual data points.

11) Our journal encourages inclusion of *data citations in the reference list* to directly cite datasets that were re-used and obtained from public databases. Data citations in the article text are distinct from normal bibliographical citations and should directly link to the database records from which the data can be accessed. In the main text, data citations are formatted as follows: "Data ref: Smith et al, 2001" or "Data ref: NCBI Sequence Read Archive PRJNA342805, 2017". In the Reference list, data citations must be labeled with "[DATASET]". A data reference must provide the database name, accession number/identifiers and a resolvable link to the landing page from which the data can be accessed at the end of the reference. Further instructions are available at .

12) All Materials and Methods need to be described in the main text. We would encourage you to use 'Structured Methods', our new Materials and Methods format. According to this format, the Materials and Methods section should include a Reagents and Tools Table (listing key reagents, experimental models, software and relevant equipment and including their sources and relevant identifiers) followed by a Methods and Protocols section in which we encourage the authors to describe their methods using a step-by-step protocol format with bullet points, to facilitate the adoption of the methodologies across labs. More information on how to adhere to this format as well as downloadable templates (.doc or .xls) for the Reagents and Tools Table can be found in our author guidelines: <

<https://www.embopress.org/page/journal/14693178/authorguide#manuscriptpreparation>>. An example of a Method paper with Structured Methods can be found here: .

13) As part of the EMBO publication's Transparent Editorial Process, EMBO Reports publishes online a Review Process File to accompany accepted manuscripts. This File will be published in conjunction with your paper and will include the referee reports, your point-by-point response and all pertinent correspondence relating to the manuscript.

You can use this link to submit your revision: <https://embor.msubmit.net/cgi-bin/main.plex>

Yours sincerely,

Referee #1:

The manuscript submitted by Kiae Kim and Ka Young Chung entitled:

"Molecular mechanism of β -arrestin-2 interaction with phosphatidylinositol 4,5-bisphosphate"

represents very timely work and is interesting. However a few points could be improved in the current version of the manuscript to make it more easy to access for the reader.

Abstract

- In the first sentence the authors write "...bind to the N-domain of arrestin, resulting in β XX release". To make it more accessible to readers not as familiar with the topic as the authors, please consider to replace " β XX" with the more general term "C-terminus" as this is wider known.

- In the second sentence, the authors wrote "..., which transform arrestins into the active state." Please consider to change it to "..., which transform arrestins into the receptor-activated state." To specifically state once that they are referring to that. As arrestins are scaffolding proteins without enzymatic function one could argue that we know too little about their activity in general, but the focus here is on the function specifically related to receptor activation.

Figures:

- General:

- o It would be very helpful for the reader to link the original data from EV ("Expanded View"/supplement) also in figure legends, as it is done in the text, especially for the HDX-MS comparison profiles
- o for an easier understanding/comparison of the changes of barr2 dynamics, subheadings at the respective figure panels (e.g. Fig. 1D, E,...) with indication of the interaction partner (e.g. V2Rpp or PIP2) might help in the figures
- o Please show also the data and analysis for the statistics in supplement

- Fig. 1

- o A: the use of another color to highlight the polar core region (now: cyan circle) could significantly improve the visibility, as the distal C-term is also shown in cyan and they are located closely to each other
- o B: at the upper part of the model, there is the C' indicated as in Fig 1A, however the dotted C-term is missing. Either you replace this C' to the resolved part or indicate the unresolved part here as well (e.g. in gray). Further, it would be helpful to distinguish the basal and V2Rpp-bound C-terminus legend (by color or by C' and C')
- o Are the different time-points used at all? Please clarify which time-point is visualized in the figure?

- Fig. 3

- o D: The figure legend is quite difficult to read. I would suggest to re-write it partially and mention the used colors directly after

the respective region in brackets or bring at least all loops together and indicate the linker separately.

- Fig. 4

o A general legend to the scheme is missing. Suggestion: "Here bArr2 is shown schematically as a gray shape, particular regions are highlighted in dark gray (dotted) lines. Secondary structures (beta sheets) of interest are shown as arrows. Positively charged region in C-domain is indicated by red color and "+" symbols. PIP2 is shown as orange cylinder. "

Results:

- General:

o The authors compare the HDX levels often in two dimensions e.g. PIP2-bound mutated barr2 to the WT changes (e.g. Fig 2C compared to Fig. 1E) and also to the apo mut. barr2 (Fig. 2C). Here, understanding and explanations would be improved, if you are showing it as direct comparison e.g. by radar chart or as two dimensional graph with comparison of mut. vs WT bArr2 and on the other axis apo vs PIP2-bound bArr2 and/or an extra structure (as those you have done already) with HDX level changes based on the compared structures (e.g. values from Fig. 2C - Fig. 1E)

o Additionally it would be very helpful to be able to use the vast data of the authors as a resource for different moving regions in form of a table of the structural domains with indicated peptides for PIP2-binding, V2Rpp-binding, -mutants etc. A table might help to indicate regions that move differently, mutants affected or unaffected, or similar between PIP2- and V2Rpp-binding.

- To: conformational changes of barr2 upon PIP2-binding

o Barr1 was used in Figure 1A, B and C, however the HDX-MS was done with bArr2 it should be mentioned that bArr1 was used as model or you may use the new structure of barr2 with V2Rpp (Maharana et al. 2023, Mol Cell, PDB: 8GOC)

o Paragraph one: The authors write "For comparative analysis, we also examined the changes in HDX levels upon V2Rpp-binding..." Please clearly state that this was in absence of PIP2 to make it easier to follow and emphasize the difference of these conditions (not a "sequential" comparison PIP2-binding only, then PIP2 binding and V2Rpp-binding as potential model for receptor interaction).

o Paragraph three: The authors state that the "HDX-MS analyses data well-reflected the V2Rpp-induced conformational changes of bArr2". In comparison to what? Do the authors mean as expected from other studies/other methods? If yes, please cite the papers you are comparing it to.

o paragraph four: you refer to the bArr1 residues regarding the binding, did you check the amino acid sequence of bArr2 in respect to differences? So that the reason for differences in HDX levels, in the respective regions, also might be caused by distinct interaction partners?

The reason I am asking/mentioning it is that in Janetzko et al. 2022, Cell, those authors have investigated the PIP2 interaction mutants for bArr1 and 2 and the positions are just one number shifted in the sequence. But in my opinion it would be interesting for/benefiting the discussion to bring it up as a discussion point, after all the bArr isoforms seem to mediate overlapping but also distinct functions, therefore a (theoretical) comparison (based on sequences and published results) would be interesting. Especially since bArr2 seems to be more dependent on finger loop for receptor interaction, interesting question whether the PIP2-dependency/(pre-)activation could differ as well.

- To: Distal C-tail of barr2 is not involved in PIP2-induced activation

o last paragraph: You are describing that barr2 1-394 shows HDX changes in regions that did not change in WT. In the last paragraph however you mention that those were similar to WT (Figure EV3). Would you like to indicate that it is not significantly different? In this case, please write it like this and indicate significantly different changes.

- To: Y64 in the finger loop is not involved in PIP2-induced activation

o First paragraph: you are referring in the beginning only to your own results, so please indicate this by "We have shown..." or check for other studies, showing the activation induced conformational changes in finger, middle and C-loop or in N- and C-domain generally.

o End of first paragraph: "Thus, the mutation ..." is a hypothesis or are the authors referring to other work/results? Please clarify.

- To: The lariat loop of barr2 is involved in PIP2-induced activation

o First paragraph: As you are focusing on the L280 mutant in this subsection, it would improve the reading flow if you consider to explain it here generally and introduce the residues in the sections where they are discussed

o Begin of second paragraph: As this is the explanation to the first paragraph regarding the L280 mutation, you may improve the reading by indicating it here. E.g. "We have chosen the lariat loop for further examination because our initial observations (Fig. 1E...) indicate that HDX levels here were altered upon PIP2 binding..."

o End of second paragraph: a scheme of the L280G mutation and how it would affect the interactions might be helpful

o Third paragraph: Are those changes (Fig. 3B) significant? Please explain...

- To: The back loop of barr2 is involved in PIP2-induced activation

o First paragraph: how do the authors explain that the HDX-MS profiles of the back loop were not affected by the binding of PIP2 even though it apparently plays a role in PIP2-induced activation/conformational changes? Possibly more appropriate to write in the discussion.

o Second paragraph:

Here a scheme of the mutation and changes in interaction with other residues might be very helpful for the reader; also please clarify if this does refer to basal without binding to anything? Please indicate clearly to make it easier to follow.

I am a bit confused by these some statements in this section:

The authors write "Compared to the HDX levels in the WT, E315A showed higher HDX levels at the back loops... reflecting altered conformational dynamics of the back loop upon E315A mutation." While in the third paragraph however the authors start with "Although E315A did not show appreciable differences in the conformational dynamics compared to the WT, ...". It is hard to follow here what the authors refer to but for the readers it sounds contradicting.

Also please clarify if the changes in Fig. 3E are significant?? Does the first sentence of third paragraph indicate, that those differences were not significant ("Although E315A did not show appreciable differences in the conformational dynamics compared to the WT...")? Please, phrase clearly.

Discussion:

- Paragraph five: In the description of Fig. 4 you explain step 4 before step 3, however it would be also or even better understandable if you follow the actual order.

- It would be very interesting if the authors could discuss their findings in context of the recent publication by Grimes et al. 2023, Cell, as those authors reported that bArrestin can pre-associate with the membrane via the finger loop. They also investigated the bArr2-PIP interaction mutant and found that the delta PIP mutant did not influence the basal bArr2 molecules in the membrane, but changed the agonist-dependent interaction with the plasma membrane, as well as changing the dynamics of accumulation in clathrin-coated pits. Could it be a different mode of basal interaction? Also how does the extended lipid anchoring deficient mutant (Δ ELA)

(R189Q/F191E/L192S/M193G/T226S/K227E/T228S/K230Q/K231E/K233Q/R237E/K251Q/K325Q/K327Q/V329S/V330D/R332E) used by Grimes et al. correlate to the HDX-MS findings or the proposed route of allosteric conformational changes? For this mutant construct, the authors found a diminished lateral diffusion in the plasma membrane at a basal state. How can these findings be connected? Please speculate if possible

- In the beginning of the results the authors state that the deuterium change was measured for several durations (10, 100, 1000, 10000 seconds). For which duration did the authors show the results and why? Did the authors observe interesting differences? What was the expectation to measure different durations? It would be great if the authors could discuss this in the manuscript.

Material and Methods:

- To: barr2 expression and purification

o Instead of 0.03 mM 30 μ M for easiness

- To: statistical analysis

o The authors describe statistical analysis in the methods but don't indicate the results anywhere in the figures/results.

o Please show the data and analysis results in supplement and indicate where to find them

o Please show in the respective figures (or the figure legends) which changes are significant and which not, or indicating that all shown changes are significant

Referee #2:

The study of Kiae Kim and Ka Young Chung comprehensively analyzed dynamic conformational changes of the entire β -arr2 upon phosphatidylinositol 4,5-bisphosphate (PIP2)-binding using HDX-MS.

Indeed, previous investigations revealed that binding of PIP2 to the β -arr2 C-domain contributes to its activation. In this context, recent studies had monitored conformational change induced by PIP2 binding at selected regions of β -arr2 using fluorescence labelling or 19F-NMR (Janetzko et al 2022; Zhai et al., 2023).

The present work provides further insight on the « triggering » mechanism for this PIP2-mediated activation of β -arr2.

The investigations with the HDX-MS approach seem well performed and the experiments are well documented and explained throughout the manuscript, allowing the non-specialists (non-structuralists) to follow the demonstration.

The major conclusion of the paper is that PIP2 binding at the C-domain of β -arr2 impacts the back loop, which destabilizes the gate loop and beta strand XX (in the C-tail) to transform β -arr2 into a pre-active state. These conclusions are based on the thorough analysis of the different exchange rates in different regions with WT and β -arr2 mutants +/- PIP2.

For a general audience journal such as EMBO reports, an experimental validation of the proposed model would be a plus. For example, the β -arr2 mutant E315K in the Eichel et al 2018 paper, (E314K in the rat sequence) promotes constitutive CCP targeting. The authors say that mutating L280 to G in the lariat loop blocks the transformation of β -arr2 into its pre-activated state (i.e. disturbance of gate loop and β XX). According to the author hypothesis, in the double mutant E314K/L280G the constitutive CCP targeting should be blocked. Of course other equivalent experiments are possible.

Also, in "real-life", the C-terminal phosphorylated tail of the activating GPCR and plasma membrane associated PIP2 binding near the receptor contribute to β -arr2 activation. In the discussion section the authors should explain to what extent separate experiments in solution with the receptor C-terminal peptide or PIP2 are representative of the physiological activation and what are potential limitations of the approach.

Finally, we also think it would be useful to give a general explanation of the structure of beta-arrestin stating how it is composed in the abstract/introduction/beginning of results section for the "non-aficionados" of arrestin structure (i.e. how many beta-strands etc.)

We greatly appreciate the time and effort the reviewers have dedicated to evaluating our manuscript. We understand the importance of addressing the comments raised and are thankful for the opportunity to clarify and enhance our study on the structural mechanisms of PIP₂-mediated arrestin activation.

Referee #1:

The manuscript submitted by Kiae Kim and Ka Young Chung entitled: "Molecular mechanism of β -arrestin-2 interaction with phosphatidylinositol 4,5-bisphosphate" represents very timely work and is interesting. However, a few points could be improved in the current version of the manuscript to make it more easy to access for the reader.

Abstract

- In the first sentence the authors write "...bind to the N-domain of arrestin, resulting in β XX release". To make it more accessible to readers not as familiar with the topic as the authors, please consider to replace " β XX" with the more general term "C-terminus" as this is wider known.

Response: It has been changed.

- In the second sentence, the authors wrote "..., which transform arrestins into the active state." Please consider to change it to "..., which transform arrestins into the receptor-activated state." To specifically state once that they are referring to that. As arrestins are scaffolding proteins without enzymatic function one could argue that we know too little about their activity in general, but the focus here is on the function specifically related to receptor activation.

Response: It has been changed.

Figures:

- General:

o It would be very helpful for the reader to link the original data from EV ("Expanded View"/supplement) also in figure legends, as it is done in the text, especially for the HDX-MS comparison profiles

Response: It is now provided in the figure legends.

o for an easier understanding/comparison of the changes of barr2 dynamics, subheadings at the respective figure panels (e.g. Fig. 1D, E,...) with indication of the interaction partner (e.g. V2Rpp or PIP₂) might help in the figures

Response: Thank you for the suggestion. We have attempted to show the interaction partners. However, the available structures with the interaction partners often do not show the C-terminus of barr2. Thus, we used basal state structure to illustrate the HDX-MS data. To help the readers better understand the binding interfaces, we have provided structures with interaction partners in Fig. 1B and 1C.

o Please show also the data and analysis for the statistics in supplement

Response: The detailed data can be found in Dataset EV1, and in the figure legends we stated that the analysis was done by Student's t-test.

- Fig. 1

o A: the use of another color to highlight the polar core region (now: cyan circle) could significantly improve the visibility, as the distal C-term is also shown in cyan and they are located closely to each other

Response: The circle is now colored with orange.

o B: at the upper part of the model, there is the C' indicated as in Fig 1A, however the dotted C-term is missing. Either you replace this C' to the resolved part or indicate the unresolved part here as well (e.g. in gray). Further, it would be helpful to distinguish the basal and V2Rpp-bound C-terminus legend (by color or by C' and C')

Response: The figure is now revised as the reviewer suggested.

o Are the different time-points used at all? Please clarify which time-point is visualized in the figure?

Response: The color-coded HDX level differences are based on the maximum differences at any D₂O incubation time point. Now, we have stated this in the figure legends as below.

“The color-coded HDX level differences are based on the maximum differences at any D₂O incubation time point.”

- Fig. 3

o D: The figure legend is quite difficult to read. I would suggest to re-write it partially and mention the used colors directly after the respective region in brackets or bring at least all loops together and indicate the linker separately.

Response: The figure legend is now revised as the reviewer suggested.

- Fig. 4

o A general legend to the scheme is missing. Suggestion: "Here bArr2 is shown schematically as a gray shape, particular regions are highlighted in dark gray (dotted) lines. Secondary structures (beta sheets) of interest are shown as arrows. Positively charged region in C-domain is indicated by red color and "+" symbols. PIP2 is shown

as orange cylinder. "

Response: Thank you for the suggestion. The figure legend is now revised as the reviewer suggested.

Results:

- General:

o The authors compare the HDX levels often in two dimensions e.g. PIP2-bound mutated barr2 to the WT changes (e.g. Fig 2C compared to Fig. 1E) and also to the apo mut. barr2 (Fig. 2C). Here, understanding and explanations would be improved, if you are showing it as direct comparison e.g. by radar chart or as two dimensional graph with comparison of mut. vs WT bArr2 and on the other axis apo vs PIP2-bound bArr2 and/or an extra structure (as those you have done already) with HDX level changes based on the compared structures (e.g. values from Fig. 2C - Fig. 1E)

Response: Thank you for the suggestion. Unfortunately, we cannot directly analyze values of Fig. 2C – Fig. 1E because the absolute HDX levels could be slightly different when the experiments are conducted independently. Therefore, we only perform direct comparisons when the samples were prepared at the same time and when the mass spectrometric analyses were conducted at the same time.

Nevertheless, we understand the reviewer's concern, and we prepared Table EV1 as suggested in the below comment.

o Additionally it would be very helpful to be able to use the vast data of the authors as a resource for different moving regions in form of a table of the structural domains with indicated peptides for PIP2-binding, V2Rpp-binding, -mutants etc. A table might help to indicate regions that move differently, mutants affected or unaffected, or similar between PIP2- and V2Rpp-binding.

Response: As the reviewer suggested, we generated Table EV1 that summarized the effects of PIP₂-binding on the selected regions.

- To: conformational changes of barr2 upon PIP2-binding

o Barr1 was used in Figure 1A, B and C, however the HDX-MS was done with bArr2 □ it should be mentioned that bArr1 was used as model or you may use the new structure of barr2 with V2Rpp (Maharana et al. 2023, Mol Cell, PDB: 8GOC)

Response: We used bArr1 structure because there is no bArr2 structure bound to PIP₂. In the manuscript text and figure legends, it is clearly state that the structures in the figures are bArr1.

o Paragraph one: The authors write "For comparatice analysis, we also examined the changes in HDX levels upon

V2Rpp-binding..." Please clearly state that this was in absence of PIP₂ to make it easier to follow and emphasize the difference of these conditions (not a "sequential" comparison PIP₂-binding only, then PIP₂ binding and V2Rpp-binding as potential model for receptor interaction).

Response: The paragraph is now revised as below.

"To compare the structures changes of PIP₂ binding with the phosphorylated GPCR binding, we also examined the changes in HDX levels upon V2Rpp-binding without PIP₂ addition because V2Rpp is a well-established model system for understanding β arr interactions with phosphorylated receptor C-tails (Figure 1B) (Latorraca et al., 2020; Mayer et al., 2019; Shukla et al., 2013; Yang et al., 2015)."

o Paragraph three: The authors state that the "HDX-MS analyses data well-reflected the V2Rpp-induced conformational changes of β arr2". In comparison to what? Do the authors mean as expected from other studies/other methods? If yes, please cite the papers you are comparing it to.

Response: The sentence is now revised as below.

"The HDX-MS analyses data well-reflected the V2Rpp-induced conformational changes of β arr2 as illustrated in Fig 1B."

o paragraph four: you refer to the β arr1 residues regarding the binding, did you check the amino acid sequence of β arr2 in respect to differences? So that the reason for differences in HDX levels, in the respective regions, also might be caused by distinct interaction partners? The reason I am asking/mentioning it is that in Janetzko et al. 2022, Cell, those authors have investigated the PIP₂ interaction mutants for β arr1 and 2 and the positions are just one number shifted in the sequence. But in my opinion it would be interesting for/benefiting the discussion to bring it up as a discussion point, after all the β arr isoforms seem to mediate overlapping but also distinct functions, therefore a (theoretical) comparison (based on sequences and published results) would be interesting. Especially since β arr2 seems to be more dependent on finger loop for receptor interaction, interesting question whether the PIP₂-dependency/(pre-)activation could differ as well.

Response: Thank you for the comments. We have checked the sequence, and the positively charged residues are conserved. As the reviewer suggested, we revised the manuscript text as below.

"This may be due to three reasons. First, HDX monitors the buffer exposure of the amide hydrogen atoms at the peptide backbone. The binding of PIP₂ at β XV and β XVI occurs through the charge-charge interaction mediated by the amino acid side chains. If this interaction had not affected the backbone amide hydrogens, we would not have been able to detect changes in HDX levels in these regions even after ligand binding. Second, the interacting residues may differ slightly between the receptor-bound (i.e., PIP₂-bound state shown in the NTSR1- β arr1 and GCGR- β arr1 complexes) and unbound states (i.e., current study). Even in the receptor-bound states, PIP₂ interacted differently between the NTSR1-bound and GCGR-bound states (Figure 1C, inset). Third, the reported β arr structures with PIP₂ are β arr1 structures (Figure 1C), whereas in this study, we analyzed the conformation of β arr2. Therefore, the differences may stem from variations between these subtypes. Nevertheless, the HDX-MS data indicate that PIP₂ interacts at the positively charged region within the C-domain of β arr2."

- To: Distal C-tail of β arr2 is not involved in PIP₂-induced activation

o last paragraph: You are describing that barr2 1-394 shows HDX changes in regions that did not change in WT. In the last paragraph however you mention that those were similar to WT (Figure EV3). Would you like to indicate that it is not significantly different? In this case, please write it like this and indicate significantly different changes.

Response: We have revised the sentence as below.

“A few other regions where we did not observe HDX changes with PIP2-bound WT were also affected (Figure EV3, peptides 50 – 64, 118 –127, 128 – 145, and 251 – 258), but the HDX levels of these regions became statistically no different to those of the WT (Figure EV3, peptides 128 – 145 and 251 – 258) or similar to those of the WT (Figure EV3, peptides 50 – 64 and 118 –127).”

- To: Y64 in the finger loop is not involved in PIP2-induced activation

o First paragraph: you are referring in the beginning only to your own results, so please indicate this by "We have shown..." or check for other studies, showing the activation induced conformational changes in finger, middle and C-loop or in N- and C-domain generally.

Response: We described the conformational changes observed in the reported structures. To clarify this, we added references to this sentence. Moreover, we carefully examined the cited references throughout the manuscript, and revised them when it is necessary.

o End of first paragraph: "Thus, the mutation ..." is a hypothesis or are the authors referring to other work/results? Please clarify.

Response: We have revised the sentence as below.

“Thus, we reasoned that the mutation of Y64 destabilizes the interactions between these three loops and breaks off the transmission route from the PIP₂-binding sites”

- To: The lariat loop of βarr2 is involved in PIP2-induced activation

o First paragraph: As you are focusing on the L280 mutant in this subsection, it would improve the reading flow if you consider to explain it here generally and introduce the residues in the sections where they are discussed

Response: The description of the residue L280 is described in the second paragraph.

o Begin of second paragraph: As this is the explanation to the first paragraph regarding the L280 mutation, you may improve the reading by indicating it here. E.g. "We have chosen the lariat loop for further examination because our initial observations (Fig. 1E...) indicate that HDX levels here were altered upon PIP2 binding..."

Response: We have revised the sentence as the reviewer suggested.

“We have chosen the lariat loop for further examination because our initial observations of HDX levels in the lariat loop were altered upon the binding of PIP₂ (Figure 1E and EV2, peptide 281 – 291).”

o End of second paragraph: a scheme of the L280G mutation and how it would affect the interactions might be helpful

Response: Thank you for the suggestion. We believe that we provided the scheme by showing residues surrounding L280 in Figure 3A.

o Third paragraph: Are those changes (Fig. 3B) significant? Please explain...

Response: The changes were significant, and it is indicated in Fig EV4. In the revised manuscript text, we added “statistically significant” where appropriate.

- To: The back loop of β arr2 is involved in PIP₂-induced activation

o First paragraph: how do the authors explain that the HDX-MS profiles of the back loop were not affected by the binding of PIP₂ even though it apparently plays a role in PIP₂-induced activation/conformational changes? Possibly more appropriate to write in the discussion.

Response: We have added the following sentence in the first paragraph.

“The reason there is no differences in the HDX level of the back loop might be because it is exposed to the buffer in a very flexible form, thus the amide hydrogens in the back loop were not affected even after the conformation was altered upon binding of PIP₂”

o Second paragraph:

Here a scheme of the mutation and changes in interaction with other residues might be very helpful for the reader; also please clarify it this does refer to basal without binding to anything? Please indicate clearly to make it easier to follow.

Response: We have added the following sentence in the second paragraph.

“To test this hypothesis, we mutated E315 to alanine, which would break the interaction between E315 and Y78 (Figure 3D).”

I am a bit confused by these some statements in this section:

The authors write "Compared to the HDX levels in the WT, E315A showed higher HDX levels at the back loops... reflecting altered conformational dynamics of the back loop upon E315A mutation." While in the third paragraph however the authors start with "Although E315A did not show appreciable differences in the conformational dynamics compared to the WT, ...". It is hard to follow here what the authors refer to but for the readers it sounds contradicting.

Response: We are sorry about the confusion. The sentence is now revised as below.

"Although E315A did not show appreciable differences in the conformational dynamics other than the back loop and β VII/ β VIII loop compared to the WT, the effects of the binding of PIP₂ on E315A were dramatically different from those on the WT β arr2"

Also please clarify if the changes in Fig. 3E are significant?? Does the first sentence of third paragraph indicate, that those differences were not significant ("Although E315A did not show appreciable differences in the conformational dynamics compared to the WT...")? Please, phrase clearly.

Response: It was significant as shown in Figure EV5.

Discussion:

- Paragraph five: In the description of Fig. 4 you explain step 4 before step 3, however it would be also or even better understandable if you follow the actual order.

Response: Thank you for the suggestion. We stated step 4 before step 3 because we do not have experimental evidence (such as mutational studies) for step 3.

- It would be very interesting if the authors could discuss their findings in context of the recent publication by Grimes et al. 2023, Cell, as those authors reported that bArrestin can pre-associate with the membrane via the finger loop. They also investigated the bArr2-PIP interaction mutant and found that the delta PIP mutant did not influence the basal bArr2 molecules in the membrane, but changed the agonist-dependent interaction with the plasma membrane, as well as changing the dynamics of accumulation in clathrin-coated pits. Could it be a different mode of basal interaction? Also how does the extended lipid anchoring deficient mutant (Δ ELA) (R189Q/F191E/L192S/M193G/T226S/K227E/T228S/K230Q/K231E/K233Q/R237E/K251Q/K325Q/K327Q/V329S/V330D/R332E) used by Grimes et al. correlate to the HDX-MS findings or the proposed route of allosteric conformational changes? For this mutant construct, the authors found a diminished lateral diffusion in the plasma membrane at a basal state. How can these findings be connected? Please speculate if possible

Response: Thank you for the comment. In the revised manuscript, we discussed the Grimes's et al.'s study, and correlated their results with our current data.

"However, several studies have indicated that β arr interaction with PIP₂ without activated receptors is physiologically relevant. Grimes et al. demonstrated that β arr can spontaneously translocate to the plasma membrane independently of GPCR interaction (Grimes et al., 2023). They also found that while PIP₂ influences

agonist-dependent β arr translocation, it does not affect β arr's spontaneous membrane translocation. These findings imply that β arr can translocate to the plasma membrane before GPCR activation, and therefore β arr might be able to interact with PIP_2 even without the agonist-activated GPCRs."

"Additionally, we confirmed that PIP_2 binding enhances β arr2' interaction with V2Rpp, consistent with Grimes et al.'s findings that PIP_2 influences agonist-dependent β arr translocation (Grimes et al., 2023)."

Regarding the Δ ELA mutant and its effect on lateral diffusion in the plasma membrane, we did not correlate our data with those of Grimes's et al. because the current manuscript focuses on the allosteric route of β arr pre-activation by PIP_2 but does not study lateral diffusion in the plasma membrane. Lateral diffusion might also be affected by other factors such as the charge status of the phospholipids, cholesterol, etc.

- In the beginning of the results the authors state that the deuterium change was measured for several durations (10, 100, 1000, 10000 seconds). For which duration did the authors show the results and why? Did the authors observe interesting differences? What was the expectation to measure different durations? It would be great if the authors could discuss this in the manuscript.

Response: The reason for using different durations is that in some cases, conformational differences can be observed either over shorter or longer durations. The results from all the duration points are shown in Dataset EV1 and Figure EVs, and the statistical differences are also indicated in Figure EVs. In the main figures, the color-coded HDX level differences are based on the maximum differences at any D_2O incubation time point. We have now stated this in the figure legends as below.

"The color-coded HDX level differences are based on the maximum differences at any D_2O incubation time point."

Material and Methods:

- To: β arr2 expression and purification

o Instead of 0.03 mM \square 30 μM for easiness

Response: It has been changed.

- To: statistical analysis

o The authors describe statistical analysis in the methods but don't indicate the results anywhere in the figures/results.

o Please show the data and analysis results in supplement and indicate where to find them

o Please show in the respective figures (or the figure legends) which changes are significant and which not, or indicating that all shown changes are significant

Response: The statistical analyses are indicated in the Figure EV2-EV5, and the detailed data are summarized in the Dataset EV1.

Referee #2:

The study of Kiae Kim and Ka Young Chung comprehensively analyzed dynamic conformational changes of the entire β -arr2 upon phosphatidylinositol 4,5-bisphosphate (PIP₂)-binding using HDX-MS. Indeed, previous investigations revealed that binding of PIP₂ to the β -arr2 C-domain contributes to its activation. In this context, recent studies had monitored conformational change induced by PIP₂ binding at selected regions of β -arr2 using fluorescence labelling or 19F-NMR (Janetzko et al 2022; Zhai et al., 2023). The present work provides further insight on the « triggering » mechanism for this PIP₂-mediated activation of β -arr2. The investigations with the HDX-MS approach seem well performed and the experiments are well documented and explained throughout the manuscript, allowing the non-specialists (non-structuralists) to follow the demonstration.

The major conclusion of the paper is that PIP₂ binding at the C-domain of β -arr2 impacts the back loop, which destabilizes the gate loop and beta strand XX (in the C-tail) to transform β -arr2 into a pre-active state. These conclusions are based on the thorough analysis of the different exchange rates in different regions with WT and β -arr2 mutants +/- PIP₂.

For a general audience journal such as EMBO reports, an experimental validation of the proposed model would be a plus. For example, the β -arr2 mutant E315K in the Eichel et al 2018 paper, (E314K in the rat sequence) promotes constitutive CCP targeting. The authors say that mutating L280 to G in the lariat loop blocks the transformation of β -arr2 into its pre-activated state (i.e. disturbance of gate loop and β XX). According to the author hypothesis, in the double mutant E314K/L280G the constitutive CCP targeting should be blocked. Of course other equivalent experiments are possible.

Response: To validate the model, we developed an experimental system that can measure arrestin's C-tail release as an indicator of its activation. We tested the effect of PIP₂ pre-incubation on V2Rpp-induced C-tail release. Initially, we generated an E315A/L280G double mutant as suggested by the reviewer, but we were unable to purify this construct due to constant aggregation. Therefore, we used L280G to examine whether disrupting the allosteric conformational change route at the gate loop affects the impact of PIP₂ pre-incubation on V2Rpp-induced C-tail release. The details are described below.

“PIP₂ facilitates V2Rpp-induced β arr2 activation

A previous study reported that the simultaneous binding of V2Rpp and PIP₂ induces complex conformational changes in different structural regions (Zhai et al., 2023). Here, we tested whether pre-incubation with PIP₂ affects the V2Rpp-induced C-tail release. To examine the C-tail release of β arr2, we developed an experimental system using bimane fluorophore, an environment-sensitive fluorescent molecule. Bimane fluorescence can be quenched by nearby tryptophan residues (Jones Brunette & Farrens, 2014). We placed bimane at the β arr2 C-tail (residue 385) and tryptophan at the nearby N-terminus (residue 6) (Figure 4A). We generated cysteine-free β arr2 (Cys-free β arr2: C17S/C60A/C126S/C141I/C151V/C244V/C253V/C271S/C395S/C410S) to prevent monobromobimane labeling at the unwanted sites, substituted aspartate at residue 385 with cysteine (D385C), and substituted glycine at residue 6 to tryptophan (G6W). In the basal state, bimane fluorescence is quenched by the nearby the tryptophan residue (Figure 4A, upper panel). Upon release of the C-tail, quenching is abolished as residue 385 moves away from residue 6 (Figure 4A, lower panel).

When we incubated the bimane-labeled β arr2 with excess V2Rpp (300 μ M), bimane fluorescence increased (Figure 4B). To examine the pre-activation effect of PIP₂, we reduced V2Rpp concentration to 30 μ M, where it induces minimal bimane fluorescence increase (Figure 4C). Similarly, 30 μ M PIP₂ did not induce C-tail release (Figure 4C). However, pre-incubation with PIP₂ followed by V2Rpp addition significantly increased the bimane fluorescence (Figure 4C), implying that PIP₂ pre-activation facilitates V2Rpp-induced β arr2 activation.

Introducing the L280G mutation to disrupt the allosteric conformational pathway reduced the augmentation of the V2Rpp-induced C-tail release after PIP₂ pre-incubation (Figure 4C). This result suggests that the lariat loop is the allosteric conformational change route through which PIP₂ binding facilitates phosphorylated receptor-induced β arr2 activation.”

Also, in "real-life", the C-terminal phosphorylated tail of the activating GPCR and plasma membrane associated PIP₂ binding near the receptor contribute to β -arr2 activation. In the discussion section the authors should explain to what extent separate experiments in solution with the receptor C-terminal peptide or PIP₂ are representative of the physiological activation and what are potential limitations of the approach.

Response: Thank you for the suggestion. In the revised manuscript, we discussed that, within the cell, arrestins might be able to interact with PIP₂ without the receptor.

“However, several studies have indicated that β arr interaction with PIP₂ without activated receptors is physiologically relevant. Grimes et al. demonstrated that β arr can spontaneously translocate to the plasma membrane independently of GPCR interaction (Grimes et al., 2023). They also found that while PIP₂ influences agonist-dependent β arr translocation, it does not affect β arr’s spontaneous membrane translocation. These findings imply that β arr can translocate to the plasma membrane before GPCR activation, and therefore β arr might be able to interact with PIP₂ even without the agonist-activated GPCRs. Additionally, Eichel et al. reported that β arr can accumulate at the clathrin-coated endocytic structure (CCS) even after dissociating from agonist-activated GPCRs, with PIP₂ being crucial for this accumulation (Eichel et al., 2018).”

We further discussed the limitations of this study in the following paragraph.

“Although this study highlights the potential routes for PIP₂-induced pre-activation of β arr2, it comes with limitations. First, within the cell, arrestins interact with a variety of other components, including phospholipids, receptors, G proteins, and signaling proteins such as ERK1/2 (Chen et al., 2023; Grimes et al., 2023; Lally et al., 2017; Qu et al, 2021b; Smith et al, 2021). Therefore, the conformational changes observed in the purified β arr2 upon interaction with PIP₂ or V2Rpp might be too simplistic compared to the complex nature within the cell. Second, our study couldn't detail the allosteric conformation changes at the atomic level due to technical constraints. Advances in biophysical techniques, such as time-resolved Cryo-EM (Klebl et al, 2023), could provide deeper insights into the step-by-step conformational changes at the atomic level.”

Finally, we also think it would be useful to give a general explanation of the structure of beta-arrestin stating how it is composed in the abstract/introduction/beginning of results section for the "non-aficionados" of arrestin structure (i.e. how many beta-strands etc.)

Response: We have modified a sentence in Introduction as below.

“These studies showed that arrestins consist of N- and C-domains with a seven-stranded β sandwich in each domain (Figure 1A). The two domains are connected by one linker between β X and β XI, and the C-terminal part including β XX, which is located in the N-domain in the basal state, is connected by a flexible linker to the C-domain (Figure 1A)”

Dear Prof. Chung

Thank you for the submission of your revised manuscript to EMBO reports. We have now received the full set of referee reports that is copied below.

As you will see, both referees are very positive about the study and request only minor changes, i.e., the supplementation of Table EV1. That said, I noted that you have uploaded a 'Table EV1' so it might have been overlooked?

From the editorial side, there are also a few things that we need before we can proceed with the official acceptance of your study.

- Your manuscript will be published in our Reports section. For short reports, the revised manuscript should not exceed 27,000 characters (including spaces but excluding materials & methods and references) and 5 main plus 5 expanded view figures. The results and discussion sections must further be combined, which will help to shorten the manuscript text by eliminating some redundancy that is inevitable when discussing the same experiments twice.

- Regarding the Author Contributions, we now use CRediT to specify the contributions of each author in the journal submission system. Therefore, please remove the Author Contributions from the manuscript file and make sure that the author contributions in our manuscript tracking system are correct and up-to-date. The information you specified in the system will be automatically retrieved and typeset into the article. You can enter additional information in the free text box provided, if you wish.

- Please upload the Expanded View figures as individual files and remove the legends from the figure itself. It is sufficient to have the legends in the manuscript file.

- Dataset EV1: is the first tab the legend for the datasets shown in the other 4 tabs? If not, please note that we need a legend for the dataset in a separate tab. Please also correct the callouts to Dataset EV1 instead of 'Data 1a' and make it clear that these descriptions relate to the different tabs and to which one in the .xls file.

- You cited Seyedabadi et al, 2021 as [PREPRINT] but it appears to have been published in the journal Biomolecules. If so, please correct the citation and remove the [PREPRINT] label from the reference list.

- Please remove the Reagents and Tools table from the manuscript and upload it as separate .docx file choosing the file type 'Reagents and Tools Table' in the online manuscript tracking system. Thank you.

- Data availability section: please include a link that resolves directly to the deposited dataset (PXD049391) and don't forget to remove the reviewer access.

- Data availability section: you state that "The proteins structures were obtained from Protein Data Bank (PDB ID: 1G4R, 3P2D, 6UP7, 8JRU, 8JRV, and 8GOC)." This seems to refer to published structures that you downloaded and used for comparison. If so, please remove this reference from the Data Availability section, as this section is meant to list only newly generated structures, i.e., those you report on in your study. Instead, please cite structures you downloaded and re-used in the text (where applicable) and in the reference list using our Data Citation format. Here, you cite both, the dataset with its URL or DOI and the paper reporting it. The data citation is labeled with the prefix "Data ref:" in the text and with [DATASET] in the reference list. Please find further information and examples in our Guide to Authors

<https://www.embopress.org/page/journal/14693178/authorguide#referencesformat>

- Materials and Methods should be Methods

- The text "Expanded View for this article is available online." is not necessary and should be removed from the manuscript.

- Our production/data editors have asked you to clarify several points in the figure legends (see below). Please incorporate these changes in the manuscript and return the revised file with tracked changes with your final manuscript submission.

a) Please note that the exact p values are not provided in the legends of figures 4c; EV 2; EV 3; EV 4; EV 5.

b) Please note that information related to n is missing in the legend of figure 4c.

c) Please note that the error bars are not defined in the legend of figure 4c.

- I introduced a few minor changes to the Abstract. Please see the edited version below my signature.

- Your manuscript provides evidence how PIP2 binding activates arrestin, doesn't it? What about using the following title: "Molecular mechanism of β -arrestin-2 activation by phosphatidylinositol 4,5-bisphosphate" or is this an overstatement?

- Finally, EMBO Reports papers are accompanied online by

A) a short (1-2 sentences) summary of the findings and their significance,
B) 2-3 bullet points highlighting key results and
C) a schematic summary figure that provides a sketch of the major findings (not a data image).
Please provide the summary figure as a separate file in PNG or JPG format at a size of 550x300-600 pixels (width x height).
Please note that the size is rather small and that text needs to be readable at the final size. Please send us this information along with the revised manuscript.

- On a different note, I would like to alert you that EMBO Press offers a new format for a video-synopsis of work published with us, which essentially is a short, author-generated film explaining the core findings in hand drawings, and, as we believe, can be very useful to increase visibility of the work. This has proven to offer a nice opportunity for exposure in particular for the first author(s) of the study. Please see the following link for representative examples and their integration into the article web page:
https://www.embopress.org/video_synopses
<https://www.embopress.org/doi/full/10.15252/emj.2019103932>

With kind regards,

Referee #1:

I would like to thank the authors for their excellent work on addressing all point I raised. I have not more concerns except that I did not find the table EV1 in the merged manuscript file as mentioned in the response letter.
Response: As the reviewer suggested, we generated Table EV1 that summarized the effects of PIP2-binding on the selected regions.
I might have overlooked it but it should be made sure that this bit of information should be added but not counted as a second revision which is not allowed for EMBO reports.

Referee #2:

The authors satisfactorily addressed all the points I raised in my previous review. Overall, the quality of the manuscript was significantly improved by the new experiments and suggested text corrections; the manuscript became more readable for non-specialists in the structural aspects of beta-arrestin activation.

Abstract

Phosphorylated residues of G protein-coupled receptors bind to the N-domain of arrestin, resulting in the release of its C-terminus. This induces further allosteric conformational changes, such as polar core disruption, alteration of interdomain loops, and domain rotation, which transform arrestins into the receptor-activated state. It is widely accepted that arrestin activation occurs by conformational changes propagated from the N- to the C-domain. However, recent studies have revealed that binding of phosphatidylinositol 4,5-bisphosphate (PIP2) to the C-domain transforms arrestins into an active state. Here, we aimed to elucidate the mechanisms underlying PIP2-induced arrestin activation. We compare the conformational changes of β -arrestin-2 upon binding of PIP2 or phosphorylated C-tail peptide of vasopressin receptor type 2 using hydrogen/deuterium exchange mass spectrometry (HDX-MS). Introducing point mutations on the potential routes of the allosteric conformational changes and analyzing these mutant constructs with HDX-MS reveals that PIP2-binding at the C-domain affects the back loop, which destabilizes the gate loop and β XX to transform β -arrestin-2 into the pre-active state. -

All editorial and formatting issues were resolved by the authors.

Prof. Ka Young Chung
Sungkyunkwan University
School of Pharmacy, Sungkyunkwan University
2066 Seobu-ro, Jangan-gu
School of Pharmacy, room 530525
Suwon, Gyeonggi-do 16419
Korea, Democratic People's Republic of

Dear Prof. Chung,

My colleague Martina is out of office. Therefore, I have stepped in as the secondary handling editor of your manuscript. Thank you for submitting your revised manuscript. I have now looked at everything and all is fine. Therefore, I am very pleased to accept your manuscript for publication in EMBO Reports.

Congratulations on a nice work!

Kind regards,

Deniz Senyilmaz Tiebe

--

Deniz Senyilmaz Tiebe, PhD
Senior Scientific Editor
EMBO Reports

--
